# Samples Are Not Equal: A Sample Selection Approach for Deep Clustering

**Zhengxing Jiao[1], Yaxin Hou[1], Jun Ma[1], Yuhang Li[1], Ding Ding[1], Yuheng Jia[1,2,3],***
**Hui Liu[3], Junhui Hou[4]**

[1]School of Computer Science and Engineering, Southeast University, Nanjing 210096, China
[2]Key Laboratory of New Generation Artificial Intelligence Technology and Its
Interdisciplinary Applications (Southeast University), Ministry of Education, China
[3]School of Computing Information Sciences, Saint Francis University, Hong Kong, China
[4]Department of Computer Science, City University of Hong Kong, Hong Kong, China
`{jiaozx,yaxin,220242351,yuhangli,dingding-1,yhjia}@seu.edu.cn,`
`h2liu@sfu.edu.hk, jh.hou@cityu.edu.hk`

## Abstract

Deep clustering has recently achieved remarkable progress across various domains. However, existing clustering methods typically treat all samples equally, neglecting the inherent differences in their feature patterns and learning states. Such redundant learning often drives models to overemphasize simple feature patterns in high-density regions, weakening their ability to capture complex yet diverse ones in low-density regions. To address this issue, we propose a novel plug-in designed to mitigate overfitting to simple and redundant feature patterns while encouraging the learning of more complex yet diverse ones. Specifically, we introduce a density-aware clustering head initialization strategy that adaptively adjusts each sample's contribution to cluster prototypes according to its local density in the feature space. This strategy mitigates the bias towards high-density regions and encourages a more comprehensive attention on medium- and low-density ones. Furthermore, we design a dynamic sample selection strategy that evaluates the learning state of samples based on the feature consistency and pseudo-label stability. By removing sufficiently learned samples and prioritizing unstable ones, this strategy adaptively reallocates training resources, enabling the model to consistently focus on samples that remain under-learned throughout training. Our method can be integrated as a plug-in into a wide range of deep clustering architectures. Extensive experiments on multiple benchmark datasets demonstrate that our method improves clustering accuracy by up to **6.1%** and enhances training efficiency by up to **1.3×**. Code is available at
https://github.com/notoaudrey/Samples-Are-Not-Equal

## 1 Introduction

Deep clustering integrates the representation learning ability of deep neural networks with the common objective used in traditional clustering algorithms, achieving significant advances in recent years (Zhou et al., 2025; Ren et al., 2025). Existing deep clustering methods utilize self-supervised learning techniques (He et al., 2020; Chen et al., 2020a; Grill et al., 2020) to handle complex high-dimensional data effectively, thus mitigate the limits of traditional clustering algorithms (Xie et al., 2016; Zhang et al., 2025).

Recent deep clustering methods (Gansbeke et al., 2020; Li et al., 2021; Jia et al., 2025) use pretrained encoders to obtain discriminative representations. A key property of these representations is that adjacent representations often share semantic similarity. Consequently, these methods exploit this underlying structure to generate supervisory signals for training. To further investigate the representation distribution in the feature space, we analyze the local density using the $k$-nearest neighbor distance and observe a consistent phenomenon across datasets, most samples concentrated in

---

*Corresponding author.

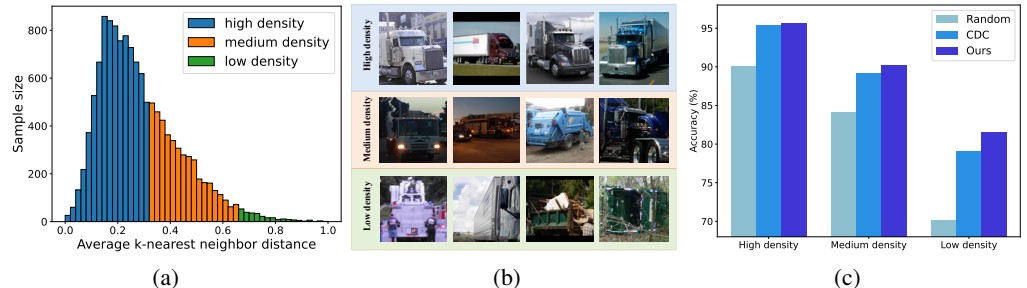

Figure 1: **(a)** Sample size distribution across varying densities in the STL-10 dataset, with density estimated by $k$-nearest neighbor distance. **(b)** Random selected images from different density regions of the STL-10 dataset. High-density samples are more simple and redundant, while medium- and low-density samples are more complex and diverse. **(c)** Performance across different density regions under three initialization strategies (i.e., Random, CDC, and Ours), showing our proposed method's consistent improvements, particularly in low-density samples.

high-density regions, while medium- and low-density regions contain far fewer samples (Fig. 1(a)). Moreover, images selected from different density regions reveal distinct feature patterns. High-density samples tend to be simple and redundant, while medium- and low-density samples exhibit greater complexity and diversity (Fig. 1(b)). For example, in the STL-10 truck class, high-density samples usually represent typical truck appearances, capturing views from the side or oblique front with holistic features. In contrast, medium- and low-density samples present richer variations, covering various truck models, multiple viewpoints, and fine-grained local details. As illustrated in Fig. 1(c), such differences in feature patterns lead to notable performance disparities, simple and redundant high-density samples are learned quickly due to their abundance and ease, while more complex and diverse medium- and low-density samples remain difficult to master. We attribute this phenomenon to the overfitting of simple and redundant feature patterns.

In this paper, we introduce a novel plug-in designed to mitigate overfitting to simple and redundant feature patterns while encouraging the learning of more complex and diverse ones. Our method tackles this challenge from two perspectives. First, we propose a density-aware clustering head initialization strategy, which adaptively weights each sample's contribution to its cluster prototype based on local density, thus reducing the bias caused by simple and redundant high-density samples dominating the clustering head initialization. Second, we develop a dynamic sample selection strategy guided by feature consistency and pseudo-label stability, which identifies and temporarily suspends training on already well-learned samples. This strategy adaptively reallocates the model's learning capacity toward more unstable samples, thereby fostering a more efficient and comprehensive learning trajectory.

The proposed method is **plug-and-play**, enabling seamless integration with existing deep clustering methods while significantly improving their clustering performance and training efficiency. In summary, the main contributions of this work are as follows.

- We identify a general phenomenon in deep clustering where sample distributions are dominated by simple and redundant samples in high-density regions. This dominance causes models overfitting to simple and redundant samples and overlook complex and diverse samples in low-density regions, limiting their discriminative power.

- We propose two strategies: (i) density-aware clustering head initialization employs local density based adaptive weighting to prevent cluster prototype bias towards high-density regions; and (ii) feature consistency and pseudo-label stability based dynamic sample selection identifies and temporarily discards sufficiently learned samples during training, enabling models to allocate more capacity to unstable samples.

- Extensive experiments on multiple benchmark datasets show that our method consistently boosts the performance of state-of-the-art deep clustering methods by up to **6.1%**. Moreover, it improves training efficiency by up to **1.3×**, achieving superior results with fewer training samples.

## 2 RELATED WORK

**Deep Clustering**. Deep clustering leverages the representation learning capability of deep neural networks to perform clustering. Existing methods can be broadly divided into two categories. The first paradigm consists of two-stage methods, where a deep neural network is first trained to extract low-dimensional representations, which are then clustered using traditional algorithms such as K-Means or spectral clustering (Xie et al., 2016; Huang et al., 2014). While this paradigm simplifies optimization, its performance is inherently constrained by the quality of extracted representations, which may not be optimally aligned with the final clustering objective. The second paradigm encompasses end-to-end iterative methods that jointly optimize representations and cluster assignments. These methods utilize the model's own outputs as supervisory signals and can be further distinguished by their learning strategy: self-training iteratively generates pseudo-labels for all samples to guide optimization (Xie et al., 2016; Guo et al., 2017); self-labeling further selects high-confidence predictions to provide more stable and reliable supervisory signals (Gansbeke et al., 2020; Jia et al., 2025; Wu et al., 2025; Li et al., 2025a); and contrastive learning enhances representations by managing similarity relationships across different data views (Shen et al., 2021; Li et al., 2021; Li & Jia, 2025; Li et al., 2025b). Recently, methods that leverage large-scale pre-trained models like CLIP have emerged as a powerful approach. By utilizing pseudo-labels derived from textual representations or enforcing cross-modal neighborhood consistency, these techniques introduce rich external knowledge, leading to significant performance gains (Cai et al., 2023; Li et al., 2024; Qiu et al., 2024). Overall, due to their effective exploitation of high-confidence samples, end-to-end self-labeling methods represent a particularly promising direction in deep clustering.

**Sample Selection**. Sample selection has been widely studied in deep learning for enhancing training efficiency, model robustness, and generalization. Existing methods can be broadly classified as either static or dynamic. Static methods perform an one-time sample selection, either prior to or during the initial stage of training, to construct a compact yet representative subset. For example, data pruning techniques eliminate redundant or low-value samples based on metrics such as forgetting events (Toneva et al., 2019; Killamsetty et al., 2021c) or influence functions (Paul et al., 2021), while core-set methods (Xia et al., 2023; Braverman et al., 2022) seek to identify a minimal subset that best approximates the full data distribution. Although effective at reducing computational cost, a key limitation of static methods is their inability to adapt to the evolving importance of samples as training progresses. In contrast, dynamic methods continuously adjust sample weights or sampling probabilities throughout the training process to prioritize the most informative or challenging examples (Nguyen et al., 2023; Yuan et al., 2025). A prominent dynamic technique is importance sampling, where a sample's selection probability is proportional to its estimated utility. Common utility metrics include loss values (i.e., emphasizing hard examples), gradient norms (i.e., quantifying a sample's contribution to parameter updates), and prediction uncertainty (i.e., identifying ambiguous samples near decision boundaries) (Chang et al., 2017a; Mindermann et al., 2022; Killamsetty et al., 2021b). By adaptively focusing learning resources on such informative data, these dynamic mechanisms promote more efficient training and improved model generalization.

**Active Learning**. Sampling strategies are also a central topic in Active Learning (AL), where the goal is to improve model performance with limited annotation budgets. AL methods select informative samples based on criteria such as uncertainty (Gal et al., 2017) and diversity or representativeness (Sener & Savarese, 2018) . Recent work further shows that sample importance should be updated dynamically as the model changes (Ash et al., 2020; Killamsetty et al., 2021a).

However, most existing sample selection methods are designed for supervised learning. These strategies become inapplicable in the unsupervised deep clustering paradigm due to the absence of ground-truth labels. To bridge this gap, we introduce a novel dynamic sample selection strategy specifically tailored for deep clustering. Our method enables the model to adaptively discard sufficiently learned samples throughout the training process, enhancing both clustering performance and training efficiency in a fully unsupervised manner. Although AL focuses on selecting samples for labeling, the idea of estimating how "valuable" each sample is is related to our setting. In deep clustering, no labels are available, and our goal is different: instead of querying informative samples, we aim to remove samples that have already been well learned so the model can focus on more challenging ones.

## 3 PROPOSED METHOD

**Overview.** Our proposed deep clustering enhancement component explicitly incorporates sample-level diversity modeling during both the clustering head initialization and training phases. This approach is designed to counteract learning bias caused by imbalanced sample feature patterns. The method consists of two core components: a density-aware clustering head initialization strategy and a dynamic sample filtering strategy based on feature consistency. In the initialization phase, we use a pre-trained feature encoder to obtain preliminary feature representations (Sec. 3.1). These representations are then used to execute our density-aware clustering head initialization strategy, which aims to generate a set of initial cluster prototypes not dominated by high-density, redundant samples. During the iterative training phase (Sec. 3.2), we adopt a dynamic training strategy. We use the feature consistency-based filtering strategy to identify and temporarily remove samples that have become stable. This allows the model to focus more learning resources on complex or unstable samples that have not been fully learned. Notably, our method is designed as a plug-in that can be readily incorporated into a wide range of deep clustering architectures, consistently improving their clustering performance. We present the pseudo-code of our method in Appendix D.

**Notation.** Denote by $\mathcal{D}_u = \{x_i : i \in \{1, 2, \ldots, N\}\}$ a training set of $N$ unlabeled samples belonging to $K$ semantic clusters. A deep clustering model is composed of a feature encoder $f_\theta(\cdot)$ and a clustering head $g(\cdot)$, where $\theta$ denotes the network parameters of the feature encoder. The feature encoder maps an input sample to a high-dimensional feature representation $z = f_\theta(x)$, while the clustering head predicts the pseudo-label distribution $p = g_\phi(z)$, where $\phi$ denotes the parameters of the clustering head. For each sample $x_i$, we apply weak augmentation $\Omega^w$ and strong augmentation $\Omega^s$ to get two different views $x_i^w$ and $x_i^s$, and their predictive distributions are denoted as $P_i^w$ and $P_i^s$.

### 3.1 DENSITY-AWARE CLUSTERING HEAD INITIALIZATION

Most clustering-head-based methods leverage self-supervised learning to obtain a pre-trained feature encoder and then attach a clustering head for further training. However, the parameters of the clustering head are typically initialized randomly (Gansbeke et al., 2020; Li et al., 2021), which is unstable and often disrupts the pre-trained representations, thereby slowing convergence and reducing robustness. To address this, CDC (Jia et al., 2025) introduces prototype-based initialization, which mitigates the degradation caused by random initialization. Nevertheless, despite its stability, prototype-based initialization struggles with imbalanced feature distributions. Since it computes cluster prototypes by averaging all sample features, the resulting prototypes are inevitably dominated by the abundant, redundant samples in high-density regions, failing to capture the true data structure. Consequently, the initial prototypes biased toward high-density regions hinder the model from effectively learning diverse feature patterns.

To mitigate this issue, we propose a density-aware clustering head initialization strategy to adaptively adjust each sample's contribution to the initial prototypes according to its local density in the feature space (Fig. 2(b)). The procedure consists of two key steps: (1) perform an initial clustering assignment using features extracted by a pre-trained backbone; (2) recompute the cluster prototypes via a density-weighted aggregation within each cluster.

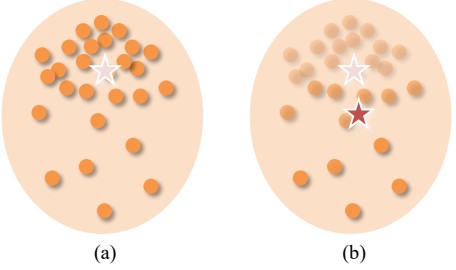

(a)          (b)

☆ Original Prototype ★ Desity-weighted Prototype

Figure 2: **(a)** Cluster prototypes dominated by high-density samples. **(b)** Density-aware initialization, where sample weights are adaptively adjusted according to density, with color intensity indicating weight magnitude.

Specifically, we begin by extracting feature representations for all samples using the pre-trained encoder $f_\theta(\cdot)$, resulting in $\mathcal{Z} = \{z_1, \ldots, z_N\}$. We then perform an initial clustering by applying the standard K-Means algorithm on $\mathcal{Z}$ to obtain pseudo-labels $\mathcal{L} = \{l_1, \ldots, l_N\}$. Based on these assignments, we introduce density-aware re-estimation of cluster prototypes. For each sample, we first estimate its local density using the average $k$-nearest

neighbor distance within the same cluster. The corresponding local density weight $w_i$ is defined as:

$$w_i = \exp\left(\alpha \cdot \frac{1}{k} \sum_{j=1}^{k} \|z_i - z_i^{(j)}\|_2\right), \tag{1}$$

where $z_i^{(j)}$ denotes the $j$-th nearest neighbor of $z_i$ within its assigned cluster, and $\alpha$ is a tunable hyperparameter controlling the sensitivity of the weights to distance. Larger values of $\alpha$ make the weights more sensitive to changes in local density. Once the density weights are computed, the prototype $c_j$ for cluster $j$ is obtained by the weighted average:

$$c_j = \frac{\sum_{z_i \in C_j} w_i \, z_i}{\sum_{z_i \in C_j} w_i}, \tag{2}$$

where $C_j$ denotes the set of samples assigned to cluster $j$. This density-aware re-estimation enhances the contribution of low-density samples while attenuating the dominance of high-density ones. Finally, the refined prototypes $\mathcal{C} = \{c_1, \ldots, c_K\}$ are employed to initialize the clustering head weights of the deep clustering model. By incorporating density information, this strategy better preserves cluster structures across regions of varying density and mitigates bias toward redundant samples, thereby providing a stronger starting point for subsequent deep clustering training.

## 3.2 DYNAMIC SAMPLE SELECTION

In unsupervised clustering tasks, models tend to quickly learn to group samples from high-density regions, which are often characterized by simple and redundant patterns. Continuing to allocate computational resources to these well-learned samples is inefficient and may hinder effective learning of samples from low-density regions, which are typically more complex and information-rich. We initially considered a density-based sample removal strategy, where local density is estimated via $k$-nearest neighbor distance to discard redundant samples in high-density regions. However, this approach is computationally expensive due to repeated neighbor searches and global distance computations, and density estimates in high-dimensional feature spaces may be imprecise, potentially leading to the unintended removal of informative samples and limiting model generalization. To address this challenge, we propose a dynamic sample selection strategy that adaptively allocates learning resources based on each sample's learning state during training. Unlike supervised learning, clustering lacks explicit ground-truth labels, making it difficult to assess sample difficulty. Traditional sample difficulty measures based on loss or gradient dynamics often fail to accurately reflect the actual learning state of samples. Although pseudo-labels offer an intuitive alternative, they are prone to noise, particularly in the early stages of training, which risks introducing bias into sample selection.

Our method builds upon the principle of prediction consistency. The central idea is that a well-learned sample should yield stable and robust feature representations, such that the model produces consistent predictions even when the input undergoes different transformations or perturbations. In other words, if the model has developed a reliable understanding of a sample, its predicted distribution should remain largely invariant under reasonable augmentations. This property not only reflects the stability of the learned representation but also serves as an implicit indicator of the sample's learning state. By leveraging prediction consistency, we can more effectively distinguish between samples that are already well captured by the model and those that require further attention, thereby guiding dynamic resource allocation during training.

**Prediction Consistency Between Weak and Strong Views**. For each input sample $x_i$, we generate two augmented versions: a weakly augmented view $x_i^w$ and a strongly augmented view $x_i^s$. Passing these through the model yields their predictive distributions $P_i^w$ and $P_i^s$. To quantify their alignment, we define the prediction consistency score $S_i$ as the cosine similarity between the two distributions:

$$S_i = \cos(P_i^w, P_i^s). \tag{3}$$

A higher value of $S_i$ indicates that the model produces consistent predictions under different levels of augmentation, suggesting that the corresponding sample is being reliably captured by the learned representation.

**Stability Evaluation via Second-Order Differences**. High consistency at a single time step does not necessarily indicate that a sample is stably learned, as even uncertain samples may occasionally

exhibit strong agreement by chance. To better capture temporal stability, we track each sample's prediction consistency over three consecutive training epochs. Let $t$ denote the current epoch index. For sample $x_i$, the second-order difference is computed across the $t$th, $(t-1)$th, and $(t-2)$th epochs as:

$$\Delta^2 S_i^{(t)} = S_i^{(t)} - 2S_i^{(t-1)} + S_i^{(t-2)}. \tag{4}$$

This metric reflects the "acceleration" of consistency changes, effectively capturing fluctuations in the learning dynamics. A sample is considered stable when

$$|\Delta^2 S_i^{(t)}| < \epsilon, \tag{5}$$

where $\epsilon$ is a predefined stability threshold. Under this condition, the sample exhibits both high prediction consistency and temporal stability, signifying that it has reached a well-learned state.

**Pseudo-Label Consistency Check**. As an additional requirement, we further impose that a sample's pseudo-labels remain consistent across the most recent three epochs. If a sample's pseudo-label changes during this period, it is regarded as unstable, even if its prediction consistency shows only minor fluctuations. This criterion helps filter out noisy or ambiguous samples that may otherwise be misclassified as stable.

Based on the above criteria, we design a dynamic sample selection strategy that consists of two steps:

- **Exclusion**. A sample is regarded as *stable* if it simultaneously satisfies the following two conditions: (i) its prediction consistency remains stable, and (ii) its pseudo-label does not change over the recent epochs. Such samples are temporarily excluded from the training batch in order to avoid redundant computation.

- **Reinclusion**. In contrast, if a sample exhibits significant fluctuations in prediction consistency (i.e., $|\Delta^2 S_i^{(t)}| \geq \epsilon$), or if its pseudo-label changes during the observation window, it is considered unstable. These unstable samples are retained in the training queue and may even be assigned higher training priority to encourage further refinement.

This strategy allows the model to adaptively allocate computational resources toward unstable samples that require further learning, while deprioritizing stable ones. As a result, it mitigates the bias introduced by uneven feature distributions and promotes more balanced representation learning across diverse data regions. Unlike most existing sample selection strategies in supervised learning, which primarily rely on loss values or gradient magnitudes, our method is tailored for unsupervised clustering by leveraging prediction consistency and pseudo-label stability as more reliable indicators of the learning state.

## 4 EXPERIMENT

### 4.1 EXPERIMENT SETTINGS

**Datasets**. We evaluate our method on six standard benchmarks, including CIFAR-10 (Krizhevsky, 2009), CIFAR-20 (Krizhevsky, 2009), STL-10 (Coates et al., 2011), ImageNet-10 (Chang et al., 2017b), ImageNet-Dogs (Chang et al., 2017b), and Tiny-ImageNet (Le & Yang, 2015). Following previous works (Chang et al., 2017b; Jia et al., 2025), we construct ImageNet-10, ImageNet-Dogs, and Tiny-ImageNet by selecting 10, 15, and 200 subsets from ImageNet-1k (Deng et al., 2009), respectively. We strictly follow the experimental protocols of the corresponding baselines to ensure a fair and consistent comparison.

**Baselines**. We integrate our proposed plug-in into four advanced deep clustering algorithms, including CC (Li et al., 2021), TCL (Li et al., 2022), SCAN (Gansbeke et al., 2020), and CDC (Jia et al., 2025). In addition, we compare the enhanced versions against several representative deep clustering methods, including BYOL (Grill et al., 2020), NNM (Dang et al., 2021), GCC (Zhong et al., 2021), IDFD (Tao et al., 2021), TCC (Shen et al., 2021), SPICE (Niu et al., 2022), ProPos (Huang et al., 2023), SeCu (Qian, 2023), and CoNR (Yu et al., 2023).

**Experiment Settings**. We evaluate clustering performance using three standard metrics, i.e., Accuracy (ACC) (Li & Ding, 2006), Normalized Mutual Information (NMI) (Strehl & Ghosh, 2002),

and Adjusted Rand Index (ARI) (Hubert & Arabie, 1985). Higher scores indicate better alignment with the ground-truth clustering. For CC (Li et al., 2021), TCL (Li et al., 2022), SCAN (Gansbeke et al., 2020), and CDC (Jia et al., 2025), we adopt MoCo-v2 (Chen et al., 2020b) as the pre-trained backbone and strictly follow the data augmentation protocols of CDC (Jia et al., 2025).

Additional experiment settings details are provided in Appendix B.

## 4.2 MAIN RESULTS

Table 1: Comparison of clustering performance (%) on six standard benchmarks. The best result for each method is highlighted in **bold**, while the overall best result is marked with an underline.

| Method | CIFAR-10 | | | CIFAR-20 | | | STL-10 | | | ImageNet-10 | | | ImageNet-Dogs | | | Tiny-ImageNet | | | Avg. |
|---|---|---|---|---|---|---|---|---|---|---|---|---|---|---|---|---|---|---|---|
| | ACC | NMI | ARI | ACC | NMI | ARI | ACC | NMI | ARI | ACC | NMI | ARI | ACC | NMI | ARI | ACC | NMI | ARI | |
| BYOL [NeurIPS'20] | 87.5 | 78.0 | 75.2 | 52.3 | 53.3 | 36.0 | 86.1 | 75.4 | 71.5 | 94.7 | 88.4 | 88.9 | 72.9 | 69.7 | 60.9 | - | - | - | - |
| NNM [CVPR'21] | 84.3 | 74.8 | 70.9 | 47.7 | 48.4 | 31.6 | 80.8 | 69.4 | 65.0 | - | - | - | - | - | - | - | - | - | - |
| GCC [ICCV'21] | 85.6 | 76.4 | 72.8 | 47.2 | 47.2 | 30.5 | 78.8 | 68.4 | 63.1 | 90.1 | 84.2 | 82.2 | 52.6 | 49.0 | 36.2 | 13.8 | 34.7 | 7.5 | 56.7 |
| IDFD [ICLR'21] | 81.5 | 71.1 | 66.3 | 42.5 | 42.6 | 26.4 | 75.6 | 64.3 | 57.5 | 95.4 | 89.8 | 90.1 | 59.1 | 54.6 | 41.3 | - | - | - | - |
| TCC [NeurIPS'21] | 90.6 | 79.0 | 73.3 | 49.1 | 47.9 | 31.2 | 81.4 | 73.2 | 68.9 | 89.7 | 84.0 | 82.5 | 59.5 | 55.4 | 41.7 | - | - | - | - |
| SPICE [TIP'22] | 91.7 | 85.8 | 83.6 | 58.4 | 58.4 | 42.2 | 92.9 | 86.0 | 85.3 | 95.9 | 90.2 | 91.2 | 67.5 | 62.7 | 52.6 | 30.5 | 44.9 | 16.1 | 68.7 |
| ProPos [TPAMI'22] | 94.3 | 88.6 | 88.4 | 61.4 | 60.6 | 45.1 | 86.7 | 75.8 | 73.7 | 96.2 | 90.8 | 91.8 | 77.5 | 73.7 | 67.5 | 29.4 | 46.0 | 17.9 | 70.3 |
| SeCu [ICCV'23] | 93.0 | 86.1 | 85.7 | 55.2 | 55.1 | 39.6 | 83.6 | 73.3 | 69.3 | - | - | - | - | - | - | - | - | - | - |
| CoNR [NeurIPS'23] | 93.2 | 86.7 | 86.1 | 60.4 | 60.4 | 44.3 | 92.6 | 85.2 | 84.6 | 96.4 | 91.1 | 92.2 | 79.4 | 74.4 | 66.7 | 30.8 | 46.1 | 18.4 | 71.6 |
| CC [AAAI'21] | 86.3 | 77.6 | 74.3 | 52.5 | 50.9 | 35.2 | 80.0 | 72.7 | 67.7 | 90.5 | 87.6 | 84.6 | 63.3 | 60.0 | 49.3 | 13.9 | 43.0 | 5.5 | 60.8 |
| CC+Ours | **89.5** | **81.3** | **79.7** | **57.4** | **57.5** | **42.3** | **91.4** | **83.2** | **82.5** | **97.2** | **93.1** | **93.9** | **66.1** | **63.3** | **52.8** | **19.1** | **46.5** | **7.6** | **66.9 (+6.1)** |
| TCL [IJCV'22] | 88.2 | 80.4 | 76.8 | 53.1 | 52.9 | 35.7 | 86.8 | 79.9 | 75.7 | 88.4 | 83.3 | 80.0 | 64.4 | 62.3 | 51.6 | 17.2 | 45.5 | 7.4 | 62.7 |
| TCL+Ours | **90.0** | **84.1** | **80.6** | **57.9** | **57.6** | **41.1** | **90.4** | **81.8** | **80.6** | **95.3** | **91.1** | **90.1** | **73.7** | **70.1** | **60.5** | **22.1** | **46.6** | **10.1** | **68.0 (+5.3)** |
| SCAN [ECCV'20] | 90.2 | 83.7 | 81.0 | 52.1 | 54.4 | 38.0 | 91.4 | 80.4 | 82.6 | 97.2 | 92.9 | 93.9 | 71.8 | 69.1 | 60.6 | 27.4 | 51.9 | 14.1 | 68.7 |
| SCAN+Ours | **92.2** | **85.6** | **84.5** | **55.4** | **57.3** | **40.2** | **92.7** | **85.1** | **84.9** | **97.8** | **94.3** | **95.3** | **76.7** | **73.9** | **67.2** | **28.7** | **52.3** | **14.9** | **71.0 (+2.3)** |
| CDC [ICLR'25] | 94.2 | 88.1 | 88.1 | 61.9 | 60.9 | 46.1 | 93.0 | 85.8 | 85.6 | 97.1 | 92.7 | 93.6 | 79.5 | 77.0 | 70.5 | 31.3 | 45.0 | 18.0 | 72.7 |
| CDC+Ours | **94.7** | **88.7** | **89.0** | **62.7** | **61.5** | **46.6** | **93.6** | **86.9** | **86.8** | **97.2** | **92.9** | **93.8** | **84.3** | **78.2** | **73.8** | **32.2** | **46.1** | **18.6** | **73.8 (+1.1)** |

**Significant Clustering Improvement**. As shown in Table 1, our method delivers consistent and substantial improvements when integrated with four representative deep clustering methods. On six benchmark datasets, it achieves performance gains of up to **6.1%** when integrated with CC (Li et al., 2021) and a significant 1.1% even when combined with the state-of-the-art method CDC (Jia et al., 2025). Furthermore, our plug-in enables CDC to achieve the highest performance on five out of the six datasets (i.e., CIFAR-10, CIFAR-20, STL-10, ImageNet-Dogs, and Tiny-ImageNet), while its integration with SCAN (Gansbeke et al., 2020) yields the best result on ImageNet-10. These findings demonstrate the effectiveness of our method across a diverse range of baselines.

**Improved Training Efficiency**. Beyond clustering performance, we evaluate the training efficiency of our method. By dynamically pruning well-learned samples, the number of samples involved in gradient updates decreases progressively. This reduction lowers computational cost without degrading clustering performance. Table 2 quantifies the average number of samples retained per epoch when integrating our strategy with SCAN (Gansbeke et al., 2020) and CDC (Jia et al., 2025) on CIFAR-10, CIFAR-20, and STL-10. The results demonstrate that our method consistently reduces the number of training samples across all settings, leading to notable training time savings. Notably, this leads to an average efficiency improvement of 1.3×, confirming that our method enhances both clustering performance and practical training efficiency.

Table 2: Comparison of training efficiency.

| Method | CIFAR-10 | CIFAR-20 | STL-10 | Avg. |
|---|---|---|---|---|
| CDC+Ours | 17.4% | 12.1% | 14.3% | 14.6% |
| SCAN+Ours | 37.7% | 52.9% | 29.9% | 40.2% |
| CC+Ours | 44.5% | 30.9% | 28.1% | 34.5% |
| Speed Up | | ~1.3× | | |

## 4.3 FURTHER ANALYSIS

**Ablation Study**. We conduct ablation studies on CIFAR-10, CIFAR-20, and STL-10 to validate the contribution of each proposed module. As summarized in Table 3, both modules consistently im-

prove performance. Specifically, the Density-Aware Clustering Head Initialization (DACHI) yields average gains of 1.2%, 1.6%, and 1.1% on CIFAR-10, CIFAR-20, and STL-10, respectively, for both CDC (Jia et al., 2025) and SCAN (Gansbeke et al., 2020). This indicates that the proposed clustering head initialization strategy better captures diverse feature patterns. The Dynamic Sample Selection (DSS) module further enhances robustness to diverse sample feature patterns, providing additional performance gains. The synergistic effect of combining DACHI and DSS achieves the best results on all datasets. For instance, on STL-10, our proposed method outperforms the original CDC and SCAN by 1.0% and 1.8%, respectively. These consistent improvements validate the effectiveness of our proposed modules in boosting clustering performance.

Table 3: Ablation study of our proposed modules on three benchmarks (i.e., CIFAR-10, CIFAR-20, and STL-10). We enhance both CDC and SCAN with Density-Aware Clustering Head Initialization (DACHI) and Dynamic Sample Selection (DSS).

| Method | CIFAR-10 | | | CIFAR-20 | | | STL-10 | | | Avg. |
|--------|------|------|------|------|------|------|------|------|------|------|
| | ACC | NMI | ARI | ACC | NMI | ARI | ACC | NMI | ARI | |
| CDC | 94.2 | 88.1 | 88.1 | 61.9 | 60.9 | 46.1 | 93.0 | 85.8 | 85.6 | 78.2 |
| +DACHI | 94.5 | 88.4 | 88.6 | 62.5 | 61.5 | 46.6 | 93.5 | 86.6 | 86.4 | $78.7_{+0.5}$ |
| +DSS | 94.5 | 88.4 | 88.7 | 62.3 | 61.4 | 46.4 | 93.3 | 86.3 | 86.0 | $78.6_{+0.4}$ |
| Ours | **94.7** | **88.7** | **89.0** | **62.7** | **61.5** | **46.6** | **93.6** | **86.9** | **86.8** | $78.9_{+0.7}$ |
| SCAN | 90.2 | 83.7 | 81.0 | 52.1 | 54.4 | 38.0 | 91.4 | 83.4 | 82.6 | 73.0 |
| +DACHI | 91.7 | 85.2 | 83.8 | 55.1 | 57.2 | 39.8 | 92.5 | 84.8 | 84.5 | $74.9_{+1.9}$ |
| +DSS | 90.5 | 83.9 | 81.4 | 53.9 | 56.4 | 39.6 | 91.9 | 84.0 | 83.4 | $73.9_{+0.9}$ |
| Ours | **92.2** | **85.6** | **84.5** | **55.4** | **57.3** | **40.2** | **92.7** | **85.1** | **84.9** | $75.3_{+2.3}$ |

**Mitigating Overfitting to Redundant Patterns.** As shown in Table 4, all methods perform well on high-density regions with relatively simple patterns, while differences become more pronounced in medium- and low-density regions that contain more complex and diverse patterns. Compared with CDC, our method maintains comparable or slightly better performance in high-density regions, while consistently achieving gains in medium- and low-density regions. For instance, on STL-10, our method improves accuracy by 1.0% in the medium-density region and 2.4% in the low-density region. These results indicate that our approach performs better on complex, low-density patterns, effectively mitigating the tendency of clustering models to overfit redundant high-density patterns and enabling better learning of diverse and informative samples. Moreover, Fig. 3(a) illustrates the training dynamics on ImageNet-Dogs. CDC rapidly fits simple high-density patterns in the early stage but soon stagnates, suggesting overfitting to redundant patterns. In contrast, our method, though slower to converge initially, continues to improve steadily in later stages and ultimately surpasses CDC by a clear margin. This training behavior demonstrates that our approach not only avoids early overreliance on simple patterns but also sustains the discovery of complex patterns in medium- and low-density regions, leading to more generalizable representations.

Table 4: Comparison of clustering accuracy (%) across densities.

| Method | CIFAR-10 | | | STL-10 | | |
|--------|------|--------|------|------|--------|------|
| | High | Medium | Low | High | Medium | Low |
| Random | 96.3 | 88.4 | 85.6 | 90.1 | 84.1 | 70.1 |
| CDC | 97.5 | 90.3 | 87.1 | 95.4 | 89.2 | 79.1 |
| Ours | **97.7** | **90.9** | **88.4** | **95.6** | **90.2** | **81.5** |

**Effect of Sample Pruning Threshold $\epsilon$.** Table 5 further investigates the effect of the sample pruning threshold on clustering performance (i.e., ACC). The pruning ratio is directly proportional to the threshold $\epsilon$. For instance, on CIFAR-10, the ratio rises from 17.4% at $\epsilon = 0.1$ to nearly 49.9% at $\epsilon = 0.5$ and a similar trend can be observed on STL-10. We find that moderate pruning ($\epsilon = 0.1 \sim 0.3$) improves ACC. This suggests that removing well-learned samples enables the model to focus on more complex and informative examples, thereby enhancing performance. In contrast, aggressive pruning ($\epsilon = 0.4 \sim 0.5$) causes performance degradation, indicating that excessive sample removal discards valuable training samples. These results demonstrate the importance of selecting an appropriate threshold $\epsilon$ to balance performance gains with training efficiency (more details can be seen in Appendix C.1).

**Parameter Sensitivity Analysis.** Fig. 3(b) shows the impact of the density weighting coefficient $\alpha$ on clustering performance. When $\alpha = 0$ (i.e., no density weighting), performance on CIFAR-10

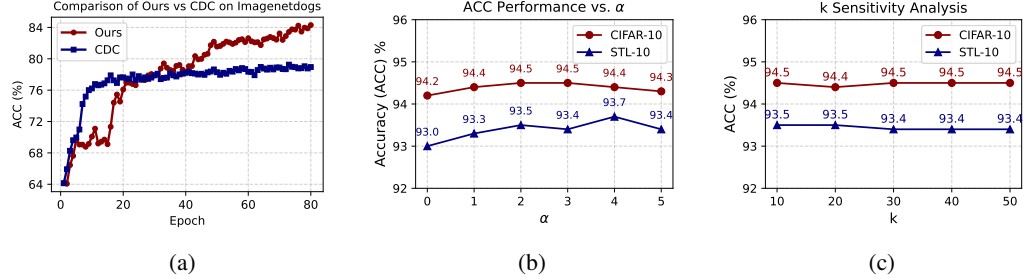

Figure 3: (**a**) Training curves on the ImageNet-Dogs dataset. (**b**) Sensitivity of clustering performance to the density weighting coefficient $\alpha$ on CIFAR-10, CIFAR-20, and STL-10. (**c**) Sensitivity of clustering performance to the number of nearest neighbors $k$ on CIFAR-10, CIFAR-20, and STL-10.

Table 5: Effect of sample pruning threshold $\epsilon$ on clustering performance.

| Dataset | $\epsilon = 0$ | | $\epsilon = 0.1$ | | $\epsilon = 0.2$ | | $\epsilon = 0.3$ | | $\epsilon = 0.4$ | | $\epsilon = 0.5$ | |
|---|---|---|---|---|---|---|---|---|---|---|---|---|
| | Pruned | ACC | Pruned | ACC | Pruned | ACC | Pruned | ACC | Pruned | ACC | Pruned | ACC |
| CIFAR-10 | 0% | 94.2 | 17.4% | 94.5 | 26.5% | 94.5 | 35.1% | 94.3 | 43.7% | 94.0 | 49.9% | 93.9 |
| STL-10 | 0% | 93.0 | 14.2% | 93.3 | 27.0% | 93.3 | 35.8% | 93.0 | 58.1% | 92.8 | 68.7% | 92.4 |

and STL-10 is relatively low. Increasing $\alpha$ to a moderate range ($\alpha = 1 \sim 2$) steadily improves clustering, achieving near-optimal results in most cases. Specifically, on CIFAR-10, ACC rises from 94.2% to 94.5%, while on STL-10, the peak ACC of 93.7% occurs at $\alpha = 4$, though further increases in $\alpha$ cause minor fluctuations. These observations suggest that moderate density weighting effectively captures diverse feature structures, whereas excessively large weights provide little additional benefit. Overall, setting $\alpha$ between 1 and 3 offers a robust and effective choice. Next, we analyze the sensitivity of our method to the choice of $k$ in density estimation, where sample density is measured by the average distance to each sample's $k$ nearest neighbors. Varying $k$ from 10 to 50, Fig. 3(c) shows that clustering performance remains highly stable, with maximum differences below 0.1% across all $k$ values. This confirms that our density estimation is robust to $k$ and does not require fine-tuning of this hyperparameter (more details can be seen in Appendix C.2).

**Our Pruning Strategy is Better.** Previous experiments demonstrated that our pruning strategy achieves better performance with higher training efficiency. To further validate its effectiveness, we compare DSS with several alternatives, including no pruning, loss-based pruning (where the second-order difference–based feature consistency loss is replaced with the standard training loss), and random pruning at different ratios(10% and 30%). As shown in Fig. 4(a), DSS consistently achieves the best results on CIFAR-10, STL-10, and the averaged scores. In contrast, loss-based pruning leads to a significant drop in performance (92.5% and 92.9%), while random pruning maintains performance at a low ratio (10%) but degrades notably as the pruning ratio increases. These results confirm that DSS is a superior pruning strategy.

**Stability Analysis of Feature Consistency**. As mentioned earlier, our method evaluates the stability of sample learning states using the second-order difference of feature consistency ($\Delta^2 S$). A lower $\Delta^2 S$ indicates smoother feature evolution and a more stable learning process. Fig. 4(b) illustrates the overall trend of $\Delta^2 S$ during training. At the early stage, the model's learning state is unstable, with $\Delta^2 S$ values remaining high. As training progresses, the model gradually learns and adapts to more samples, leading to a steady decrease in $\Delta^2 S$. Notably, CDC maintains consistently higher $\Delta^2 S$ values and continues to fluctuate in later training stages, suggesting that it overfits simple feature patterns while failing to capture more complex ones. In contrast, our method converges to a more stable state, effectively alleviating this issue. Further results in Fig. 4(c) confirm the superiority of our approach, showing consistent improvements across CIFAR-10, CIFAR-20, and STL-10.

**Discussion on Initialization Strategies**. K-Means is one of the most common strategies in clustering, and it works well on most natural image datasets. For many natural datasets, such as CIFAR-10 and STL-10, using K-Means directly can give reasonable and reliable initial centers for later clustering steps. However, the success of K-Means depends on its basic assumptions: the clusters should be roughly spherical, convex, and balanced. When a dataset does not satisfy these assumptions, K-Means may not provide a good starting point.

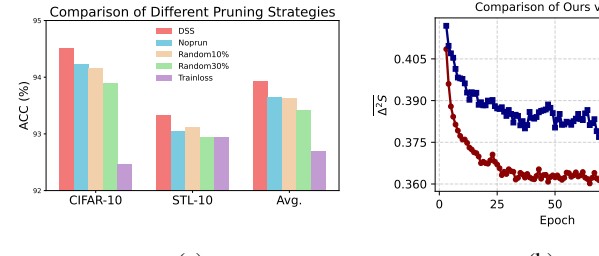 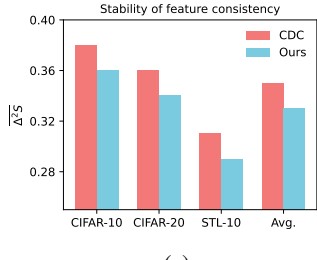

|          (a)          |          (b)          |          (c)          |

Figure 4: (**a**) Comparison of different pruning strategies. (**b**) Comparison of mean $\Delta^2 S$ on CIFAR-10, CIFAR-20, and STL-10. (**c**) Comparison of mean $\Delta^2 S$ values on CIFAR datasets between our method and CDC.

COIL-20 is a non-natural image dataset with a complex manifold structure. Its feature distribution is not spherical or convex. In this case, K-Means initialization cannot capture the true structure of the data and may lead to suboptimal clustering results. Therefore, we explored a more suitable initialization method for this type of data. Based on the distribution of COIL-20, we use spectral clustering as the initialization strategy. Spectral clustering builds a similarity graph and performs clustering in a low-dimensional space, which makes it more suitable for datasets with complex geometric shapes. After obtaining the spectral embedding, we apply our density-weighted center correction to the final K-Means step, so the initial centers better match the true data structure.

The results in Table 6 show that spectral clustering performs much better than K-Means on COIL-20, and adding our module on top of it gives a further improvement. Moreover, the results further show that choosing an appropriate initialization strategy for different data distributions is important. They also support the generality of our approach: as long as the initialization fits the data well, our density-aware correction module can further improve the clustering quality.

Table 6: Performance of different initialization strategies on COIL-20.

| Method | ACC | NMI | ARI |
|---|---|---|---|
| SCAN | 92.6 | 95.2 | 90.3 |
| SCAN + K-Means | 82.7 | 92.1 | 77.3 |
| SCAN + Spectral Clustering | 99.2 | 98.9 | 98.4 |
| SCAN + Spectral Clustering + Ours | **99.3** | **99.0** | **98.6** |

## 5 CONCLUSION

In this paper, we presented a novel plug-in designed to mitigate overfitting to simple and redundant feature patterns, which adaptively adjusts training based on sample density and learning states. Analysis of pre-trained features shows that high-density regions contain many redundant samples, while low-density regions have fewer, more diverse samples. To address this issue, we incorporated neighborhood density information into clustering head initialization, effectively reducing bias from redundant high-density samples and enhancing representations of complex low-density samples. Additionally, we introduced a dynamic sample selection strategy, defining a stability measure based on prediction consistency between weakly and strongly augmented views to prioritize unstable or under-learned samples. Extensive experiments across multiple datasets and baselines demonstrate that our method consistently outperforms existing approaches, achieving smaller fluctuations in prediction consistency and significantly improved clustering accuracy, especially on samples with complex patterns where conventional methods often struggle.

### ACKNOWLEDGMENTS

This work was supported in part by the National Natural Science Foundation of China under Grants U24A20322, 62576094 and 62422118, in part by the Hong Kong UGC under grants UGC/FDS11/E03/24, UGC/FDS11/E03/25, and in part by the Hong Kong Research Grants Council under Grant 11219324. This research work was also supported by the Big Data Computing Center of Southeast University.

ETHICS STATEMENT

We confirm that our work on deep clustering adheres to the ICLR Code of Ethics. The research utilizes publicly available datasets containing no personally identifiable information. As a strictly technical contribution focused on algorithmic improvement, we foresee no ethical issues or potential for misuse.

REPRODUCIBILITY STATEMENT

The complete code for implementing data preprocessing, model training, and evaluation is provided in the Supplementary Material to facilitate the reproduction of our results. Also, our experimental setup is described in detail in the paper. The basic experimental setup and dataset-specific hyperparameters for reproducing the results in Sec. 4.2 detailed in Appendix B.

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

# Appendix

## A  STATEMENT ON THE USE OF LARGE LANGUAGE MODELS (LLMs)

The authors used the LLM solely to assist with grammar, spelling, and sentence clarity. The authors reviewed and bear full responsibility for all content generated by the LLM. The LLM contributed in no other way to the paper. The research idea, experiment design, and all other content were developed and completed by the authors.

## B  MORE EXPERIMENT SETTINGS DETAILS

**Datasets**. Following (Gansbeke et al., 2020), we construct the CIFAR-20 dataset using 20 superclasses of the CIFAR-100 dataset, and build the ImageNet-10, ImageNet-Dogs, and Tiny-ImageNet datasets from 10, 15, and 200 classes of ImageNet-1k, respectively. For Tiny-ImageNet, we perform the complete training and testing process on the training set, and adopt the merged datasets for both training and

Table 7: Summary of datasets.

| Dataset | # Samples | # Classes | Image Size |
|---|---|---|---|
| CIFAR-10 | 60,000 | 10 | 32×32×3 |
| CIFAR-20 | 60,000 | 20 | 32×32×3 |
| STL-10 | 13,000 | 10 | 96×96×3 |
| ImageNet-10 | 13,000 | 10 | 224×224×3 |
| ImageNet-Dogs | 19,500 | 15 | 224×224×3 |
| Tiny-ImageNet | 100,000 | 200 | 64×64×3 |

testing for other datasets. Specifically, we extend the STL-10 dataset with 100,000 relevant unlabeled samples during pre-training with MoCo-v2, which are removed afterwards.

**Backbones**. We adopt ResNet-34 as the backbone in both our method and all baselines to ensure a fair comparison. To better accommodate feature extraction on small datasets such as CIFAR-10 and CIFAR-20, we replace the first convolutional filter (7×7, padding 3, stride 2) with a 3×3 filter (padding 2, stride 1), and remove the first max-pooling layer following (Huang et al., 2023; Jia et al., 2025). Besides backbone network, we attach a clustering head following the original design of each baseline method to encode the learned representation to cluster assignments.

**Experiment Settings**. For representation learning, we strictly follow the experimental settings from CDC (Jia et al., 2025) to pre-train the backbone with MoCo-v2. For data augmentation protocols, we use strong and standard augmentation from SCAN (Gansbeke et al., 2020). We adopt the adaptive moment estimation (adam) optimizer, and use learning rate from settings from CDC (Jia et al., 2025). We set the number of training epochs to 100 epochs for all datasets. Meanwhile, in all experiments that integrated our method, we set $\alpha = 2.0$ and $k = 10$ across all datasets. We set $\epsilon = 0.01$ on STL-10 and ImageNet-10 when applying our method to CC (Li et al., 2021), while $\epsilon = 0.1$ for all other cases. All experiments are conducted on an NVIDIA RTX 3090 GPU.

**Baselines**. To ensure fair comparison, we re-implement SCAN (Gansbeke et al., 2020), CC (Li et al., 2021), TCL (Li et al., 2022) and CDC (Jia et al., 2025) using the same backbone pre-trained with MoCo-v2 following the experimental settings of CDC, and we follow the original design of the clustering heads from each baseline method. We directly copy the results reported by other deep clustering methods, including BYOL (Grill et al., 2020), NNM (Dang et al., 2021), GCC (Zhong et al., 2021), IDFD (Tao et al., 2021), TCC (Shen et al., 2021), SPICE (Niu et al., 2022), Pro-Pos (Huang et al., 2023), SeCu (Qian, 2023) and CoNR (Yu et al., 2023).

## C  MORE EXPERIMENT

### C.1  EFFECT OF PRUNING THRESHOLD $\epsilon$

Table 8: Effect of pruning threshold $\epsilon$ on pruned samples and clustering performance (ACC, NMI, ARI).

| Dataset | $\epsilon = 0.1$ | | | | $\epsilon = 0.2$ | | | | $\epsilon = 0.3$ | | | | $\epsilon = 0.4$ | | | | $\epsilon = 0.5$ | | | |
|---|---|---|---|---|---|---|---|---|---|---|---|---|---|---|---|---|---|---|---|---|
| | Pruned | ACC | NMI | ARI | Pruned | ACC | NMI | ARI | Pruned | ACC | NMI | ARI | Pruned | ACC | NMI | ARI | Pruned | ACC | NMI | ARI |
| CIFAR-10 | 17.4% | 94.5 | 88.4 | 88.7 | 26.5% | 94.5 | 88.2 | 88.5 | 35.1% | 94.3 | 88.1 | 88.2 | 43.7% | 94.0 | 87.5 | 87.6 | 49.9% | 93.9 | 87.1 | 87.3 |
| STL-10 | 14.2% | 93.3 | 86.3 | 86.0 | 27.0% | 93.3 | 86.3 | 86.1 | 35.8% | 93.0 | 85.9 | 85.6 | 58.1% | 92.8 | 85.6 | 85.2 | 68.7% | 92.4 | 84.7 | 84.3 |

## C.2 SENSITIVITY ANALYSIS

Table 9: Sensitivity analysis of $\alpha$ in density weighting on CIFAR-10, CIFAR-20, and STL-10.

| Dataset | $\alpha = 0$ | | | $\alpha = 1$ | | | $\alpha = 2$ | | |
|---------|------|------|------|------|------|------|------|------|------|
| | ACC | NMI | ARI | ACC | NMI | ARI | ACC | NMI | ARI |
| CIFAR-10 | 94.2 | 88.1 | 88.1 | 94.4 | 88.2 | 88.4 | 94.5 | 88.4 | 88.6 |
| CIFAR-20 | 61.9 | 60.9 | 46.1 | 62.5 | 61.5 | 46.6 | 62.5 | 61.5 | 46.6 |
| STL-10 | 93.0 | 85.8 | 85.6 | 93.3 | 86.3 | 86.0 | 93.5 | 86.6 | 86.4 |

| Dataset | $\alpha = 3$ | | | $\alpha = 4$ | | | $\alpha = 5$ | | |
|---------|------|------|------|------|------|------|------|------|------|
| | ACC | NMI | ARI | ACC | NMI | ARI | ACC | NMI | ARI |
| CIFAR-10 | 94.5 | 88.4 | 88.6 | 94.4 | 88.3 | 88.5 | 94.3 | 88.1 | 88.2 |
| CIFAR-20 | 62.4 | 61.4 | 46.5 | 62.4 | 61.4 | 46.5 | 62.5 | 61.3 | 46.5 |
| STL-10 | 93.4 | 86.4 | 86.2 | 93.7 | 86.9 | 86.8 | 93.4 | 86.4 | 86.2 |

**Sensitivity Evaluation of Clustering Performance to Density Weighting**. As shown in Table 9, we further investigate the sensitivity of clustering performance to the density weighting coefficient $\alpha$. When $\alpha = 0$ (i.e., without density weighting), the clustering performance across all three datasets is relatively lower. As $\alpha$ increases to a moderate range (i.e., $\alpha = 1 \sim 2$), the clustering performance improves steadily, reaching an optimum in most cases. For instance, on CIFAR-10, ACC increases from 94.2% to 94.5% and NMI from 88.1% to 88.4%. On STL-10, the peak ACC of 93.7% is achieved at $\alpha = 4$, but the clustering performance suffers from minor fluctuations with a further increase in $\alpha$. This indicates that moderate density weighting effectively captures diverse feature structures, while excessively large weights yield no additional gains. In summary, a value of $\alpha$ between 1 and 3 represents a robust and effective setting at most cases.

**Sensitivity of Density Estimation to the Choice of $k$**. Our method estimates sample density using the average distance to its $k$ nearest neighbors. To assess the sensitivity to $k$, we vary it from 10 to 50 and evaluate the clustering performance on CIFAR-10, CIFAR-20, and STL-10. As shown in Fig. 5, the clustering performance across all metrics (i.e., ACC, NMI, and ARI) remains highly stable. For example, on CIFAR-10, ACC fluctuates only between 94.4% and 94.5%, while variations in NMI and ARI are within 0.2%. This stability is consistent across CIFAR-20 and STL-10, with maximum performance differences below 0.2% across all $k$ values. These results demonstrate that our density estimation is robust to the choice of $k$, indicating that the method's performance does not depend on fine-tuning this hyperparameter.

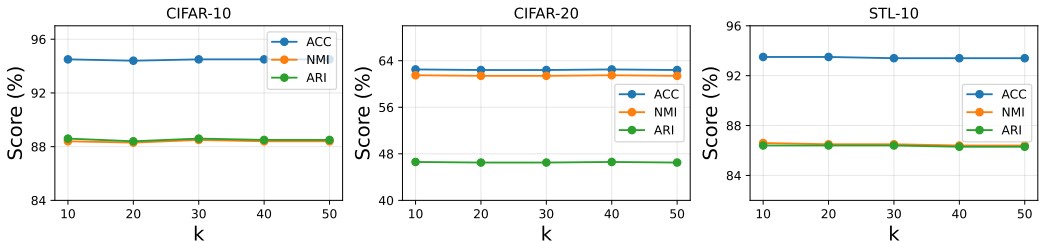

Figure 5: Sensitivity of clustering performance to the number of nearest neighbors $k$ on CIFAR-10, CIFAR-20, and STL-10.

### C.3 ACCURACY OF OVERALL SAMPLES AND REMOVED SAMPLES

We evaluated CDC+Ours on CIFAR-10 and STL-10, comparing the per-epoch accuracy of removed samples with the accuracy over all samples, as shown in Fig. 6. The results show that removed samples consistently have higher accuracy at each stage, indicating that the model usually makes correct predictions when marking samples as "stable".

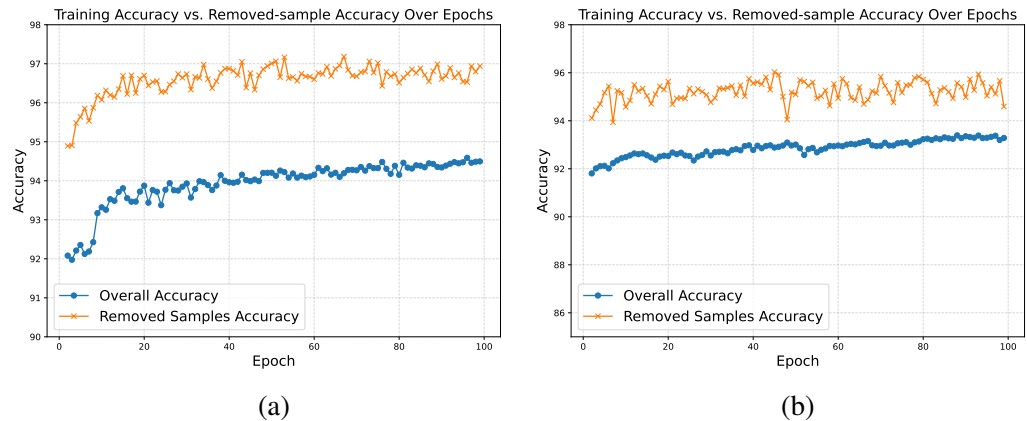

(a)                                         (b)

Figure 6: (**a**) Training accuracy and removed-sample accuracy curves on CIFAR-10 over epochs. (**b**) Training accuracy and removed-sample accuracy curves on STL-10 over epochs.

### C.4 VISUALIZATION OF SEVERAL STABLE (REMOVED) AND UNSTABLE (KEPT) SAMPLES

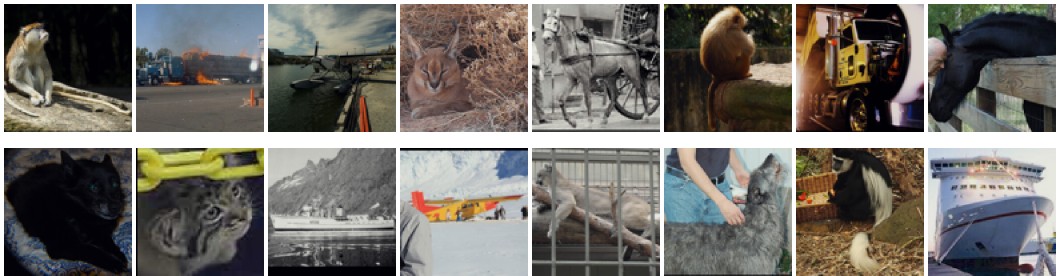

Figure 7: Visualization of some **unstable** samples from STL-10.

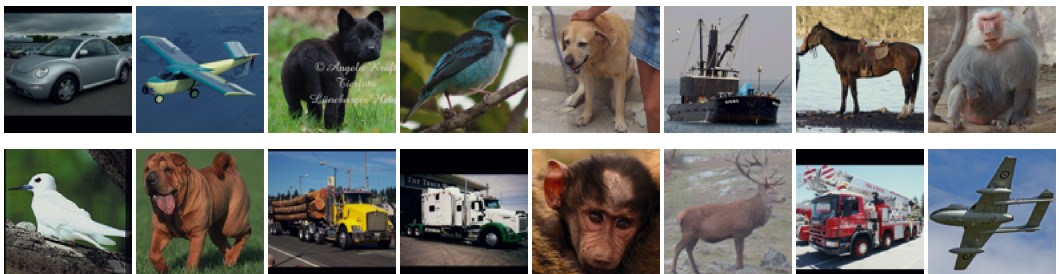

Figure 8: Visualization of some **stable** samples from STL-10.

### C.5 ADDITIONAL RESULTS UNDER MULTIPLE RANDOM SEEDS

To further evaluate the stability of our method, we conducted experiments with five different random seeds on CIFAR-10, CIFAR-20, and STL-10. The results reported in Table 10 show that our method consistently improves clustering performance across all metrics (ACC, NMI, ARI) and across all datasets. Compared with the baseline methods (SCAN and CDC), the performance gains of our approach remain stable under different random initializations, with low standard deviations.

Table 10: Clustering performance ACC, NMI, ARI (mean±std %) on CIFAR-10, CIFAR-20, and STL-10 over five runs with different random seeds.

| Method | CIFAR-10 | | | CIFAR-20 | | | STL-10 | | |
|---|---|---|---|---|---|---|---|---|---|
| | ACC | NMI | ARI | ACC | NMI | ARI | ACC | NMI | ARI |
| SCAN | 88.6±1.9 | 83.2±0.3 | 79.4±1.3 | 51.0±1.0 | 53.9±0.9 | 37.1±0.7 | 91.4±0.5 | 83.5±0.6 | 82.7±0.8 |
| SCAN+Ours | **91.8±0.5** | **85.0±0.6** | **83.7±1.0** | **55.3±0.4** | **56.7±0.5** | **40.5±0.7** | **92.5±0.1** | **85.0±0.3** | **84.6±0.3** |
| CDC | 94.1±0.3 | 87.9±0.4 | 87.8±0.6 | 61.7±0.3 | 61.1±0.2 | 46.0±0.2 | 93.0±0.1 | 85.9±0.1 | 85.6±0.1 |
| CDC+Ours | **94.6±0.2** | **88.5±0.2** | **88.7±0.4** | **62.4±0.3** | **61.4±0.1** | **46.3±0.3** | **93.5±0.1** | **86.7±0.2** | **86.6±0.3** |

## C.6 EARLY STOPPING STRATEGY

We further conducted experiments showing that our sample stability assessment can indeed serve as a practical heuristic for early stopping. Empirically, as training progresses, the number of samples that the model has "mastered" and therefore temporarily removes keeps increasing. This growth gradually slows down and eventually becomes stable. Based on this observation, we designed a simple but effective stopping rule: we track the number of removed samples at each epoch and record the maximum removed count. When this maximum remains unchanged for the most recent $K$ epochs, we stop training.

We evaluated our proposed stopping strategy in combination with CDC on CIFAR-10 and STL-10, testing several values of K(10, 20, 30, 40). We report the stopping time, the performance at the stopping point, and the best achievable performance. As shown in Table 11, for K = 10, 20, the performance at stopping is within 1% of the best performance. For K = 30, 40, the stopped model achieves performance almost identical to the optimal result. This shows that our stopping strategy can find a point that is very close to the best result without training for all epochs.

Table 11: Performance of the sample stability based early stopping strategy on CIFAR-10 and STL-10.

| | CIFAR-10 | | | | STL-10 | | | |
|---|---|---|---|---|---|---|---|---|
| | Stop Epoch | ACC | NMI | ARI | Stop Epoch | ACC | NMI | ARI |
| Epoch=100 | – | 94.4 | 88.3 | 88.5 | – | 93.4 | 86.5 | 86.2 |
| Best | – | 94.6 | 88.5 | 88.7 | – | 93.5 | 86.6 | 86.7 |
| K=10 | 49 | 94.0 | 87.2 | 87.4 | 59 | 93.0 | 85.8 | 85.5 |
| K=20 | 69 | 94.3 | 87.9 | 88.2 | 77 | 93.1 | 85.9 | 85.7 |
| K=30 | 69 | 94.3 | 87.9 | 88.2 | 98 | 93.4 | 86.6 | 86.4 |
| K=40 | – | – | – | – | 98 | 93.4 | 86.6 | 86.4 |

## C.7 EXPERIMENTS ON NON-IMAGE DATA

To further evaluate the generality of our method, we conduct additional experiments on three non-image datasets: CNAE-9, Semeion, and News20. These datasets cover diverse modalities, including high-dimensional sparse TF-IDF vectors and binarized digit patterns. We apply our module on top of SCAN and report the results in Table 12. Across all datasets, our method consistently improves over the SCAN baseline, indicating the effectiveness of our approach. These results show that our module is not tied to any specific data type and can generalize well to non-image modalities.

Table 12: Generality of our method across different data modalities.

| | CNAE-9 | | | Semeion | | | News20 | | |
|---|---|---|---|---|---|---|---|---|---|
| | ACC | NMI | ARI | ACC | NMI | ARI | ACC | NMI | ARI |
| SCAN | 55.9 | 43.6 | 34.8 | 54.0 | 46.8 | 33.4 | 60.6 | 59.6 | 46.5 |
| SCAN+Ours | **63.9** | **51.7** | **43.3** | **55.8** | **52.3** | **40.8** | **62.5** | **60.2** | **47.5** |

# D  ALGORITHM

We present the pseudo-code of our proposed method in Algorithm 1.

---

**Algorithm 1:** The proposed algorithm

---

1: **Input**: Unlabeled training data $\mathcal{D}_u = \{x_i : i \in \{1, 2, \ldots, N\}\}$.
2: **Output**: Encoder $f_\theta(\cdot)$, clustering head $g(\cdot)$, and $K$ clusters.
3: Load the pre-trained parameters of $f_\theta(\cdot)$.
4: Extract features using $f_\theta(\cdot)$ and perform K-Means.
5: Compute density weight by Eq. (1) and re-estimate density-weighted prototypes using Eq. (2).
6: Initialize clustering head parameters with the proposed density-weighted prototypes.
7: Initialize an empty set of stable samples $\mathcal{D}_s$ and an updated set of retained samples $\mathcal{D}_t = \mathcal{D}_u$.
8: **for** epoch=1, 2, … **do**
9:    **for** each sample $x_i \in \mathcal{D}_u$ **do**
10:       Forward pass to get $P_i^w$ and $P_i^s$ and compute $S_i$ using Eq. (3);
11:       Check second-order difference using Eq. (5) and check pseudo-label consistency;
12:       **if** prediction consistency is stable **and** pseudo-labels are consistent **then**
13:          Add $x_i$ to $\mathcal{D}_s$;
14:       **end if**
15:    **end for**
16:    Update model parameters using samples in $\mathcal{D}_t = \mathcal{D}_u - \mathcal{D}_s$;
17: **end for**

---

