# OpenReview forum: "Samples Are Not Equal: A Sample Selection Approach for Deep Clustering"
_ICLR.cc/2026/Conference — ICLR 2026 Poster_

### Official Review · Reviewer_K4YF · 2025-10-27

**Soundness:** 2
**Presentation:** 4
**Contribution:** 3
**Rating:** 4
**Confidence:** 3

**Summary:**

The paper identifies a key problem in deep clustering: existing methods treat all samples equally, causing them to overfit to simple, redundant feature patterns found in high-density regions of the feature space. To solve this, the authors propose a two-part plug-in designed to be integrated into existing deep clustering models:
* Density-Aware Clustering Head Initialization (DACHI): This strategy addresses the initialization bias. Instead of a standard prototype (which is just the average feature of all samples in a cluster and is thus dominated by high-density samples ), it computes a density-weighted prototype.
* Dynamic Sample Selection (DSS): This strategy adaptively manages training resources. It identifies and temporarily removes "sufficiently learned" samples from training batches. This allows the model to reallocate its capacity toward more "unstable" or under-learned samples.

**Strengths:**

1. The dynamic sample selection (DSS) strategy provides a novel and sensible method for curriculum learning in a fully unsupervised setting.
2. The proposed method successfully improves both clustering accuracy and training efficiency.
3. The method is evaluated as a plug-in for four different deep clustering baselines, demonstrating its general applicability. The ablation studies validate the contribution of both DACHI and DSS.

**Weaknesses:**

1. The paper claims a training speedup of 1.3, but this only accounts for the reduced batch size during the model update step. It fails to discuss the significant overhead of the selection mechanism itself.
2. The method introduces new and sensitive hyperparameters. The DSS strategy's pruning threshold, $\epsilon$, is particularly problematic. As shown in Table 5, performance is highly dependent on this value: moderate pruning ($\epsilon=0.1-0.3$) helps, but aggressive pruning ($\epsilon=0.5$) degrades performance. The paper provides no clear heuristic for setting this critical value, which was manually set to 1e-1 or 1e-2 depending on the dataset.
3. The 6.1% gain on CC is large, the improvement on the most recent and powerful baseline, CDC, is a more modest 1.1% average.
4. The paper repeatedly uses the term "overfitting"  to describe the model's bias toward high-density samples. This is a slightly imprecise use of the term.

**Questions:**

1. Related to weakness 1, could you provide a detailed analysis of the total wall-clock time per epoch, including the significant computational and memory overhead introduced by DSS? Specifically, what is the cost of calculating and storing prediction consistency histories for all $N$ samples at every epoch? The DACHI requires a k-nearest neighbor search within each initial cluster. How does this initialization step scale with very large datasets and high-dimensional features?
2. Related to weakness 2. Is there a more adaptive or heuristic-based method to set this threshold?
3. Have you investigated the interplay between your two modules? e.g. does the improved DACHI initialization lead to a larger or smaller set of samples being pruned by DSS later in training?

---

> ### Author Response · Authors · 2025-11-22
>
> Thank you for your valuable comments.
>
> ------
>
> **Question 1(a) & Weakness 1: Could you provide a detailed analysis of the total wall-clock time per epoch, including the significant computational and memory overhead introduced by DSS? Specifically, what is the cost of calculating and storing prediction consistency histories for all N samples at every epoch?**
>
> **Response**: We apologize for not explaining this clearly in the original manuscript. The reported training time in our paper already includes the cost of the dynamic sample selection (DSS) module. In practice, DSS introduces very little extra computation or memory usage. Although DSS adds a small cost for tracking and updating sample scores, it also **reduces the number of samples used for training at each step**. As a result, the **overall training time becomes shorter**. We provide the detailed overhead on CIFAR-10 and STL-10 in Tables Q1–Q3.
>
> In terms of time, DSS adds only 1.5 s per epoch on STL-10 (\~6.3% of the CDC baseline) and 8.4 s per epoch on CIFAR-10 (\~3.6% of the baseline). A further decomposition shows that consistency checking accounts for <1%, memory updates for \~4–5%, and the remaining cost comes primarily from the model's forward pass. Overall, the computational overhead of our method is lightweight and negligible.
>
> For memory, we store historical information using a sparse structure (a dictionary plus a fixed-length deque of size 3). Each sample retains at most three past labels and three loss values, resulting in only a small additional memory footprint.
>
> **Table Q1. Runtime and acceleration effect of DSS on CIFAR-10 and STL-10.**
> |**Time usage per epoch**|**STL-10**|**CIFAR-10**|
> |:-:|:-:|:-:|
> |CDC|23.9|232.4|
> |DSS|1.5|8.4|
> |CDC+Ours|19.2|185.8|
>
> **Table Q2. Breakdown of DSS runtime per epoch on CIFAR-10 and STL-10.**
> |**Time usage per epoch**|**STL-10**|**CIFAR-10**|
> |:-:|:-:|:-:|
> |Data loading + GPU copy|0.16|0.10|
> |Forward|1.22|7.88|
> |Consistency calculation|0.01|0.05|
> |Sample tracker update|0.07|0.36|
>
> **Table Q3. Additional memory usage (MB) per epoch of DSS on CIFAR-10 and STL-10.**
> |**Memory usage perepoch**|**STL-10**|**CIFAR-10**|
> |:-:|:-:|:-:|
> |DSS|31.4|91.5|

---

> ### Author Response · Authors · 2025-11-22
>
> **Question 1(b): The DACHI requires a k-nearest neighbor search within each initial cluster. How does this initialization step scale with very large datasets and high-dimensional features?**
>
> **Response**: The DACHI module is also lightweight in both time and memory. We support this claim through **theoretical analysis** and **experimental evidence**.
>
> **Theoretical complexity**. Let the total number of samples be $N$, feature dimension $D$, number of classes $C$, and $n_c$ the number of samples in class $c$ $(\sum n_c=N)$. Let $n_{max}⁡=max⁡_c n_c$ denote the largest cluster size, and $k$ the number of nearest neighbors. Since we perform $KNN$ **within each class** rather than globally, using sklearn’s $KNN$ implementation yields:
> - Time complexity:： $O(N^2D/C)$
> - Memory cost (storing class-wise features and distance matrices): $O(n_{max}D+n_{max}k)$
>
> This class-wise design ensures that even on large datasets (e.g., Tiny-ImageNet with 100k samples), each cluster contains only a small number of samples (≈500 for 200 classes), so the actual $KNN$ computation remains small. Since this process is executed **once at initialization**, the overall training cost is negligible.
>
> **Experimental validation.** We further measure the actual time (seconds) and memory usage (MB) on CIFAR-10, STL-10, and Tiny-ImageNet. The results, shown in Table Q4, confirm that the overhead is indeed small.
>
> **Table Q4. Measured computational and memory overhead of DACHI.**
> ||**STL-10**|**CIFAR-10**| **Tiny-ImageNet** |
> |:-:|:-:|:-:|:-:|
> |Time usage|0.79|2.24| 2.91 |
> |Memory usage|6.26|28.93| 3.79 |
>
> **Question 2 & Weakness 2: Is there a more adaptive or heuristic-based method to set $\epsilon$?**
>
> **Response**: We clarify that $\epsilon$ is **not sensitive**, and its setting is **highly consistent** across methods and datasets.
>
> First, our default value $1e^{-1}$ works well for almost all baselines and datasets. We did not tune this hyperparameter, and the performance is stable.  Only one specific scenario requires a smaller value: the CC[R1] method on STL-10 and ImageNet-10. CC uses instance-level strong/weak augmentation contrast, so the consistency between views is naturally higher. Under the same threshold, CC would prune much more data. In addition, STL-10 and ImageNet-10 contain only about 13K samples, so aggressive pruning may cause instability. Therefore, we use a slightly smaller threshold $1e^{-2}$ **only to keep the pruning rate comparable to other methods**, not because the method needs fine hyperparameter tuning.
>
> [R1] Contrastive clustering. AAAI, 2021.
>
> Second, the trend in Table 5 of the original paper(moderate pruning improves performance, heavy pruning slightly reduces performance) shows that $\epsilon$ is **controllable and adjustable**, rather than sensitive.
>
> - If the goal is **better clustering performance**, a smaller threshold can be used. Light pruning removes stable samples that the model already learns, which helps focus training on unstable samples.
> - If the goal is **training speed**, especially on large datasets, a larger threshold can be chosen. Even when pruning more than half of the samples, the performance shows only a very minor degradation. This shows that DSS is robust and works well even with aggressive pruning.
>
> In summary, $\epsilon$ has a clear **default value** ( $1e^{-1}$  in almost all cases ) and can also be **flexibly adjusted** depending on whether the user prefers higher speed or higher performance.

---

> ### Author Response · Authors · 2025-11-22
>
> **Question 3: Have you investigated the interplay between your two modules? e.g. does the improved DACHI initialization lead to a larger or smaller set of samples being pruned by DSS later in training?**
>
> **Response**: Thank you for the question. Following your suggestion, we compared three initialization strategies and their effect on the number of samples pruned by DSS: Random Initialization, CDC Initialization, and the proposed DACHI Initialization. Besides, we evaluated how different values of α in DACHI affect the number of pruned samples. The results in the Table Q5 suggest that DACHI and CDC initializations prune a similar number of samples and DACHI are not very sensitive to the $\alpha$ parameter, while Random Initialization tends to prune relatively more samples. Importantly, Random Initialization leads to noticeably lower accuracy (92.5 vs. 94.7 on CIFAR-10 and 77.8 vs. 93.7 on STL-10), which further highlights the performance advantage of our initialization strategy.
>
> **Possible reason for the small effect of $\alpha$ on pruning.** Although $\alpha$ changes the strength of density correction in DACHI, the performance improvement mainly affects low-density samples, which are usually hard to learn. Such samples may rarely reach the stability required for DSS pruning, which could explain why the overall pruning rate appears nearly unchanged.
>
> **Possible reason for higher pruning under Random Initialization.** Random Initialization lacks prior structure in the features, so the model relies more on easy and clear samples in early training. These samples might reach high prediction consistency sooner, leading DSS to prune them earlier. In contrast, DACHI provides a more structured starting point, which may allow the model to use a broader range of samples to refine features. As a result, fewer samples seem to be pruned under DACHI.
>
> **Table Q5: Effect of initialization strategies on DSS pruning rate.**
> |                       | **CIFAR-10** |                      | **STL-10** |                      |
> | --------------------- | :----------: | -------------------- | :--------: | -------------------- |
> |                       |     ACC      | 100 epoch pruned/All |    ACC     | 100 epoch pruned/All |
> | Random+DSS            |     92.5     | 11373/60000          |    77.8    | 3433/13000           |
> | CDC+DSS               |     94.5     | 10570/60000          |    93.3    | 2897/13000           |
> | DACHI($\alpha=1$)+DSS |     94.6     | 10473/60000          |    93.5    | 2833/13000           |
> | DACHI($\alpha=2$)+DSS |     94.7     | 10602/60000          |    93.6    | 2848/13000           |
> | DACHI($\alpha=3$)+DSS |     94.7     | 10462/60000          |    93.7    | 2874/13000           |
> | DACHI($\alpha=4$)+DSS |     94.7     | 10525/60000          |    93.7    | 2886/13000           |

---

> ### Author Response · Authors · 2025-11-22
>
> **Weakness 3: The 6.1% gain on CC is large, the improvement on the most recent and powerful baseline, CDC, is a more modest 1.1% average.**
>
> **Response**: Thank you for the comment. We would like to clarify that, CDC is a very strong and recent **SOTA** method, so improving CDC further is naturally difficult. In deep clustering, an improvement of around 1% on such a highly optimized and saturated SOTA baseline is usually considered meaningful and valuable. Our method still achieves an **average gain of 1.1%** on CDC across six datasets, which shows that our approach can enhance a very competitive baseline, not only weaker methods. Besides, we also tested our method on other baselines including SCAN, CC, and TCL, and all of them show consistent improvements. This further demonstrates the generality and broad applicability of our approach across different deep clustering methods.
>
> **Weakness 4: Use of "overfitting" in the manuscript**
>
> **Response**: Thank you for the comment. For the first time, we observe that existing deep clustering methods perform well on high-density regions but less well on low-density regions (see Fig.1(c) in the paper). We view this as the model focusing too much on high-density samples, and too less on the low-density samples. Accordingly, the existing deep clustering model will master too much nonessential details of the high-density samples, which is a kind of overfitting. To address this, we use two strategies: (1) density-aware correction during initialization, and (2) removing samples that have already been learned stably. Both strategies help reduce the bias toward high-density regions. This reflects a new perspective on deep clustering, from which we are also able to improve performance on different existing deep
> clustering models across different datasets. Notably, this viewpoint is also supported by other reviewers like Reviewer WmJa.

---

> ### Author Response · Authors · 2025-11-26
> **Looking forward to your further assessment**
>
> Dear Reviewer **K4YF**
>
> Thank you for taking the time to review our manuscript and for your valuable feedback and recognition. We have carefully addressed all the comments and concerns raised, as reflected in our detailed responses and the revised manuscript and supplementary material.
>
> We sincerely appreciate your efforts and look forward to your further assessment.
>
> Best regards,
>
> The Authors

---

### Official Review · Reviewer_8obk · 2025-10-28

**Soundness:** 3
**Presentation:** 3
**Contribution:** 2
**Rating:** 4
**Confidence:** 4

**Summary:**

To address the problem that existing deep clustering methods treat all samples equally, resulting in excessive focus on simple redundant features in high-density areas and neglect of complex and diverse features in low-density areas, the paper proposes a plug-in module, which includes a density-aware clustering head initialization strategy (adaptively adjusting the contribution to the clustering prototype according to the local density of the sample to reduce bias in high-density areas) and a dynamic sample selection strategy (evaluating the sample learning status based on feature consistency and pseudo-label stability, and prioritizing resources to samples that have not been fully learned); this module can be seamlessly integrated into a variety of deep clustering architectures. Experiments on benchmark datasets show that clustering accuracy improves, and the performance gain in medium- and low-density areas (complex features) is more significant.

**Strengths:**

1. The method proposed in the article is implemented as a plug-in module. It can be seamlessly integrated into a variety of mainstream deep clustering architectures without requiring significant modifications to the original model's core structure. The integration process only requires replacing the initialization step (density-aware clustering head initialization) and the embedded training loop (dynamic sample selection), thereby significantly reducing the application cost in existing systems and providing strong adaptability.
2. Based on the local density of samples (k-nearest neighbor distance), the contribution weight of samples to the cluster prototype is adaptively adjusted to reduce the dominance of high-density samples on the prototype, retain the clustering structure of medium and low-density areas (complex and diverse features), and avoid bias in the initialization stage.

**Weaknesses:**

1. Density-aware clustering head initialization relies on features extracted by a pre-trained encoder. The document uses MoCo-v2 to pre-train ResNet-34 but does not discuss the impact of the pre-trained encoder's quality on the method. If the feature representations extracted by the pre-trained encoder are of poor quality (e.g., low discrimination for complex samples), subsequent density calculations and K-Means initial clustering based on these features will be biased, thereby affecting the accuracy of cluster prototype initialization.
2. In the early stages of training, the model has not yet fully learned, and the accuracy of pseudo-labels is low. At this time, screening samples based on pseudo-label stability may mistakenly judge "complex samples that really need to be learned" as "unstable and need to be retained", resulting in unreasonable allocation of early training resources.
3. The experiment only verifies the effectiveness of the method on the image dataset, and does not involve other modal data such as text, speech, and time series. The adaptability of the method to non-image modalities has not been verified, and the applicable scenarios are limited.

**Questions:**

1. One key technique is initializing the clustering head, but this is highly dependent on the model's pre-training. How to deal with non-pre-trained models?
2. While this method improves overall training efficiency, the dynamic sample selection process requires additional computation and tracking: generating weak/strong augmentation views for each sample, calculating prediction consistency (cosine similarity), and tracking changes in second-order differences and pseudo-labels over nearly three epochs. How can this additional overhead be quantified?
3. There are fluctuations in consistency in the early stages of training. How to judge the credibility of the consistency constraints?

---

> ### Author Response · Authors · 2025-11-22
>
> Thank you for your valuable comments.
>
> ------
>
> **Question 1 & Weakness 1: Impact of pre-trained encoder on clustering head initialization**
>
> **Response**: We clarify that using a pre-trained encoder is a very common and often necessary practice in deep clustering. Since the data are entirely **unlabeled**, the model must rely on a self-supervised side task to learn meaningful representations. For this reason, representative methods, such as SCAN [R1], CDC [R2], CC [R3], ProPos[R4], and CoNR[R5], adopt either a **pre-trained encoder** or a **contrastive-learning–based representation learner** to build an initial feature space before clustering. Our method follows this widely adopted and well-established practice, rather than introducing any special requirement. The pre-trained encoder simply provides the feature space on which our density-aware correction module operates.
>
> [R1] SCAN:learningtoclassifyimageswithoutlabels. ECCV, 2020.
>
> [R2] Towards calibrated deep clustering network. ICLR, 2025.
>
> [R3] Contrastive clustering. AAAI, 2021.
>
> [R4] Learning representation for clustering via prototype scattering and positivesampling. TPAMI, 2023.
>
> [R5] Contextually affinitive neighborhood refinery for deep clustering. NeurIPS, 2023.
>
> To examine the potential influence of the pre-trained encoder, we conducted additional experiments using a smaller encoder, **ResNet-18**, which is substantially weaker than the **ResNet-34** used in the original paper. We re-trained MoCo-v2 on CIFAR-10 and STL-10, obtaining noticeably lower-quality representations that emulate a "weaker pre-training" scenario. We then evaluated CDC, CDC + ours, SCAN, and SCAN + ours under this setting.
>
> As shown in Table Q1, even with this degraded encoder, our method consistently yields stable and significant performance improvements.
>
> **Table Q1. Effect of encoder strength (ResNet-18 vs. ResNet-34) on clustering performance for CIFAR-10 and STL-10.**
> |               | **CIFAR-10** |          |          | **STL-10** |          |          |
> | :-----------: | :----------: | :------: | :------: | :--------: | :------: | :------: |
> | **ResNet-18** |      ACC     |    NMI   |    ARI   |     ACC    |    NMI   |    ARI   |
> |      CDC      |     91.1     |   83.3   |   82.3   |    74.1    |   70.2   |   62.8   |
> |    CDC+Ours   |   **91.5**   | **83.8** | **83.0** |  **74.4**  | **71.3** | **64.0** |
> |      SCAN     |     82.9     |   77.8   |   72.5   |    86.4    |   76.2   |   73.8   |
> |   SCAN+Ours   |   **84.6**   | **78.2** | **73.5** |  **87.5**  | **77.6** | **75.4** |
> |               |              |          |          |            |          |          |
> | **ResNet-34** |      ACC     |    NMI   |    ARI   |     ACC    |    NMI   |    ARI   |
> |      CDC      |     94.2     |   88.1   |   88.1   |    93.0    |   85.8   |   85.6   |
> |    CDC+Ours   |   **94.7**   | **88.7** | **89.0** |  **93.6**  | **86.9** | **86.8** |
> |      SCAN     |     90.2     |   83.7   |   81.0   |    91.4    |   83.4   |   82.6   |
> |   SCAN+Ours   |   **92.2**   | **85.6** | **84.5** |  **92.7**  | **85.1** | **84.9** |

---

> ### Author Response · Authors · 2025-11-22
>
> **Question 2: The dynamic sample selection process requires additional computation and tracking, how can this additional overhead be quantified?**
>
> **Response**: We apologize for not explaining this clearly in the original manuscript. The reported training time in our paper already includes the cost of the Dynamic Sample Selection (DSS) module. In practice, DSS introduces very little extra computation or memory usage. Although DSS adds a small cost for tracking and updating sample scores, it also **reduces the number of samples used for training at each step**. As a result, the **overall training time becomes shorter**. We provide the detailed overhead on CIFAR-10 and STL-10 in Tables Q2–Q4.
>
> In terms of time, DSS adds only 1.5 s per epoch on STL-10 (\~6.3% of the CDC baseline) and 8.4 s per epoch on CIFAR-10 (\~3.6% of the baseline). A further decomposition shows that consistency checking accounts for <1%, memory updates for \~4–5%, and the remaining cost comes primarily from the model’s forward pass. Overall, the computational overhead of our method is lightweight and negligible.
>
> For memory, we store historical information using a sparse structure (a dictionary plus a fixed-length deque of size 3). Each sample retains at most three past labels and three loss values, resulting in only a small additional memory footprint.
>
> **Table Q2. Runtime and acceleration effect of DSS on CIFAR-10 and STL-10.**
> |**Time usage per epoch**|**STL-10**|**CIFAR-10**|
> |:-:|:-:|:-:|
> |CDC|23.9|232.4|
> |DSS|1.5|8.4|
> |CDC+Ours|19.2|185.8|
>
> **Table Q3. Breakdown of DSS runtime per epoch on CIFAR-10 and STL-10.**
> |**Time usage per epoch**|**STL-10**|**CIFAR-10**|
> |:-:|:-:|:-:|
> |Data loading + GPU copy|0.16|0.10|
> |Forward|1.22|7.88|
> |Consistency calculation|0.01|0.05|
> |Sample tracker update|0.07|0.36|
>
> **Table Q4. Additional memory usage (MB) per epoch of DSS on CIFAR-10 and STL-10.**
> |**Memory usage perepoch**|**STL-10**|**CIFAR-10**|
> |:-:|:-:|:-:|
> |DSS|31.4|91.5|
>
> **Question 3 & Weakness 2: In the early stages of training, pseudo-labels and consistency signals may be unreliable, which may cause incorrect judgments and result in unreasonable allocation of early training resources. How can we judge the credibility of the consistency constraints at this stage?**
>
> **Response**: Our consistency constraints are reliable even in the early stage. Our method evaluates a sample's learning state using two strong constraints: **strong/weak augmentation consistency** (over the past three epochs) and **label consistency** (also over the past three epochs). A sample is removed only if it satisfies both constraints, meaning we measure persistent learning stability rather than momentary agreement. This design makes our method robust to prediction fluctuations in any single epoch.
>
> Moreover, the model does not start from scratch. It is first pretrained with **MoCo-v2**, yielding highly discriminative features, and our density-aware initialization strategy further provides a strong and stable starting point. Consequently, the consistency signals are reliable even in the early training stage.
>
> Table Q5 shows the accuracy of all samples compared with the removed samples during training. The results show that pseudo-label accuracy is already high at the beginning of training, and throughout the entire process, the removed samples always achieve much higher accuracy than the overall sample set. The complete training curves corresponding to these results are provided in **Appendix C.3**. These results demonstrate that the consistency signals are trustworthy from the start, and **our method does not cause any unreasonable allocation of early training resources**.
>
> We also visualized several stable (removed) and unstable (kept) samples. We found that stable samples are usually simpler and more typical, while low-consistency samples tend to be more complex. Additional examples are provided in the **Appendix C.4**.
>
> **Table Q5. Pseudo-label accuracy of overall samples and removed samples.**
> |     Epoch    |   5  |        |  10  |        |  30  |        |  50  |        |  70  |        |  100 |        |
> | :----------: | :--: | :----: | :--: | :----: | :--: | :----: | :--: | :----: | :--: | :----: | :--: | :----: |
> |              |  All | Pruned |  All | Pruned |  All | Pruned |  All | Pruned |  All | Pruned |  All | Pruned |
> | **CIFAR-10** | 92.2 |  95.5  | 93.1 |  96.2  | 93.8 |  96.6  | 94.2 |  96.9  | 94.3 |  96.7  | 94.5 |  97.1  |
> |  **STL-10**  | 92.0 |  94.7  | 92.2 |  95.2  | 92.7 |  95.1  | 93.0 |  95.2  | 93.1 |  95.2  | 93.4 |  95.4  |

---

> ### Author Response · Authors · 2025-11-22
>
> **Weakness 3: The adaptability of the method to non-image modalities has not been verified, and the applicable scenarios are limited.**
>
> **Response**: Our method generalizes well beyond image data. We tested it on several **non-image datasets** (text and tabular), and the results consistently show that our module improves clustering performance across different data modalities.
>
> Table Q6 reports the results on **CNAE-9**, **Semeion**, and **News20**, which cover diverse input types such as high-dimensional TF-IDF vectors and binarized patterns. Across all datasets, our method consistently improves over the baseline, demonstrating its robustness across different data modalities. The full results and additional analysis have been added to the Appendix C.8.
>
> **Table Q6. Generality of our method across different data modalities.**
> |               | **CNAE-9** |          |          | **Semeion** |          |          | **News20** |          |          |
> | :-----------: | :--------: | :------: | :------: | :---------: | :------: | :------: | :--------: | :------: | :------: |
> |               |     ACC    |    NMI   |    ARI   |     ACC     |    NMI   |    ARI   |     ACC    |    NMI   |    ARI   |
> |    **SCAN**   |    55.9    |   43.6   |   34.8   |     54.0    |   46.8   |   33.4   |    60.6    |   59.6   |   46.5   |
> | **SCAN+Ours** |  **63.9**  | **51.7** | **43.3** |   **55.8**  | **52.3** | **40.8** |  **62.5**  | **60.2** | **47.5** |

---

> ### Author Response · Authors · 2025-11-26
> **Looking forward to your further assessment**
>
> Dear Reviewer **8obk**
>
> Thank you for taking the time to review our manuscript and for your valuable feedback and recognition. We have carefully addressed all the comments and concerns raised, as reflected in our detailed responses and the revised manuscript and supplementary material.
>
> We sincerely appreciate your efforts and look forward to your further assessment.
>
> Best regards,
>
> The Authors

---

> > ### Comment · Reviewer_8obk · 2025-11-26
> >
> > Thank you for the author's reply. Regarding the impact of pre-trained model initialization performance on the results, the author supplemented the results with CIFAR-10 and STL-0 on ResNet-18. However, a key issue is that both datasets are relatively simple, resulting in insignificant differences between different pre-trained models. A more complex ImageNet dataset might be better suited for validating the author's proposed viewpoint.

---

> > > ### Author Response · Authors · 2025-11-27
> > >
> > > Thank you for the thoughtful feedback.
> > >
> > > We agree that CIFAR-10 and STL-10 are relatively simple datasets. However, as shown in Table Q7, the encoders pretrained with ResNet-34 and ResNet-18 show very different levels of representation quality. We use K-Means clustering accuracy to measure the quality of the pretrained features: on STL-10, the ResNet-34 encoder reaches 84% accuracy, while the ResNet-18 encoder only reaches 69%. This indicates a large gap in representation quality, which has a direct impact on model performance. As shown in Table Q7, our proposed method brings consistent and stable improvements for both encoders, indicating that our proposed method can perform well even under a poorer quality pre-trained encoder.
> > >
> > > We also agree that a more complex dataset can better validate our viewpoint. Following your suggestion, we added results on the more challenging Tiny-ImageNet dataset, as shown in Table Q7. On Tiny-ImageNet, even the ResNet-34 pretrained encoder achieves only 24% K-Means accuracy, showing the difficulty of this dataset. The results demonstrate that our method consistently improves performance on both complex datasets (Tiny-ImageNet) and simpler ones (CIFAR-10, STL-10). Moreover, when the backbone is changed from ResNet-34 to the smaller ResNet-18, the overall feature quality drops significantly, yet our method still provides stable gains, and in some cases (e.g., SCAN + Ours on Tiny-ImageNet) the improvement becomes even larger.
> > >
> > > It is also worth noting that even a weaker backbone such as ResNet-18 is still a pretrained model and already encodes similarities and distribution patterns of the data. The pretrained model will create a bias toward high-density regions. Our method directly addresses this issue by reducing such density bias and helping the model better learn from samples in low-density or harder regions. As a result, our approach remains effective even when the dataset becomes more complex or the backbone becomes weaker.
> > >
> > > **Table Q7. Effect of encoder strength (ResNet-18 vs. ResNet-34) on clustering performance(%) for Tiny-ImageNet, CIFAR-10 and STL-10.**
> > >
> > > |               | Tiny-ImageNet |          |          | **CIFAR-10** |          |          | **STL-10** |          |          |
> > > | :------------ | :-----------: | :------: | :------: | :----------: | :------: | :------: | :--------: | :------: | :------: |
> > > | **ResNet-18** |      ACC      |   NMI    |   ARI    |     ACC      |   NMI    |   ARI    |    ACC     |   NMI    |   ARI    |
> > > | CDC           |     27.2      |   42.4   |   15.3   |     91.1     |   83.3   |   82.3   |    74.1    |   70.2   |   62.8   |
> > > | CDC+Ours      |   **27.9**    | **43.1** | **15.7** |   **91.5**   | **83.8** | **83.0** |  **74.4**  | **71.3** | **64.0** |
> > > | SCAN          |     23.5      |   40.0   |   12.7   |     82.9     |   77.8   |   72.5   |    86.4    |   76.2   |   73.8   |
> > > | SCAN+Ours     |   **26.5**    | **42.1** | **14.3** |   **84.6**   | **78.2** | **73.5** |  **87.5**  | **77.6** | **75.4** |
> > > |               |               |          |          |              |          |          |            |          |          |
> > > | **ResNet-34** |      ACC      |   NMI    |   ARI    |     ACC      |   NMI    |   ARI    |    ACC     |   NMI    |   ARI    |
> > > | CDC           |     31.3      |   45.0   |   18.0   |     94.2     |   88.1   |   88.1   |    93.0    |   85.8   |   85.6   |
> > > | CDC+Ours      |   **32.2**    | **46.1** | **18.6** |   **94.7**   | **88.7** | **89.0** |  **93.6**  | **86.9** | **86.8** |
> > > | SCAN          |     27.4      |   51.9   |   14.1   |     90.2     |   83.7   |   81.0   |    91.4    |   83.4   |   82.6   |
> > > | SCAN+Ours     |   **28.7**    | **52.3** | **14.9** |   **92.2**   | **85.6** | **84.5** |  **92.7**  | **85.1** | **84.9** |

---

### Official Review · Reviewer_37QQ · 2025-10-31

**Soundness:** 3
**Presentation:** 3
**Contribution:** 4
**Rating:** 8
**Confidence:** 4

**Summary:**

The paper introduces a sample selection approach for deep clustering that addresses the problem that most clustering models treat all samples equally, even though samples in high-density regions are often redundant and simple, while those in low-density regions are complex and informative.
By relying more on samples in sparse regions (based on kNN distances), the tested models learn more effectively. The authors propose two main components:

1. Density-Aware Clustering Head Initialization (DACHI) – adjusts each sample’s weight when computing cluster prototypes, so that clusters are not dominated by redundant high-density samples.

2. Dynamic Sample Selection (DSS) – uses prediction consistency and pseudo-label stability to identify well-learned samples and temporarily remove them from training, letting the model focus on under-learned samples.

The method is plug-and-play and can be integrated into existing deep clustering algorithms.

**Strengths:**

S1) Sampling strategies have not been researched enough on in the deep clustering area. The author's method can be combined with a lot of different deep clustering methods, potentially advancing the field broadly.

S2) The methods are quite simple and intuitive.

S3) By discarding some samples in each epoch the methods are even accelerated.

**Weaknesses:**

W1) Experimental evaluation could be in more depth: how does the improvement of results depend on

a) the model complexity

b) the number / sizes of datasets

c) the number of concepts per class/cluster.

I'm missing some synthetic experiments here. Also, it would be interesting what happens for datasets with consistent densities throughout the datasets, e.g. COIL 20.
Furthermore, I would be interested in how well the methods works on, e.g., tabular data, as the paper focuses on image data.

W2) Sampling strategies are a major object of research in Active Learning. Setting the method into this context would improve the paper.

W3) How much runtime do the computations for the sampling need? Is the runtime acceleration dependent on properties of the data that can be predicted?

W4) Using k-Means as initlal clustering should be discussed more. It is not clear to me how the value of k for the initial k-Means clustering was chosen. What happens if the data does not follow typical assumptions fitting k-Means, e.g., for video data?

Especially the first three pages have quite some redundant text that could be shortened in order to tackle the above weaknesses or answer the questions below.

Minor stuff:
- Table 3 appears way before it is referred to in the text which hinders the reading flow.
- Type in line 90

**Questions:**

Q1) How does your method perform on data with consistent density, e.g. COIL20?

Q2) How does your method perform on tabular data?

Q3)  How is your sampling strategy related to Active Learning strategies? What could we learn from there, what are similarities?

---

> ### Author Response · Authors · 2025-11-22
>
> Thank you for your valuable comments.
>
> ------
>
> **Question 1 & Weakness 1(d): Performance on COIL-20.**
>
> **Response**:  We sincerely thank you for raising this very insightful question regarding datasets with consistent densities, such as **COIL-20**, which prompted us to further investigate the adaptability of our method to different data distributions.
>
> We acknowledge that our previous experiments mainly focused on natural image datasets that follow the classic K-Means assumptions, such as spherical and balanced clusters. However, COIL-20 is a dataset with a **complex manifold structure**, and its feature distribution does not follow the K-Means assumptions of **spherical or convex clusters**. When a dataset does not satisfy these assumptions, K-Means may not provide a good starting point.
>
> To better handle the non-convex and non-spherical distribution of COIL-20, we adopt **spectral clustering** as the initialization strategy. Spectral clustering constructs a similarity graph between samples and maps them to a lower-dimensional space, making the cluster structure easier to separate, and its final step typically applies K-Means in the low-dimensional space. For COIL-20, we apply our density-weighted cluster center strategy to the final K-Means step of spectral clustering. This helps obtain initial centers that better match the data distribution. We then apply our method on top of these centers, and the performance improves further. We compare the performance of different initialization strategies on COIL-20, and the results are shown in Table Q1 (Backbone: ResNet-18).
>
> **Table Q1: Performance of different initialization strategies on COIL-20.**
> |                    | **COIL20**   |          |          |
> | ------------------ | -------- | -------- | -------- |
> |                    | ACC      | NMI      | ARI      |
> | SCAN               | 92.6     | 95.2     | 90.3     |
> | SCAN+K-Means        | 82.7     | 92.1     | 77.3     |
> | SCAN+Spectral Clustering     | 99.2     | 98.9     | 98.4     |
> | SCAN+Spectral Clustering+Ours | **99.3** | **99.0** | **98.6** |
>
> We can see that K-Means initialization performs poorly on COIL-20, while spectral clustering provides a significant improvement. Applying our module on top of the spectral clustering initialization further improves performance slightly. These results show that **choosing an appropriate initialization strategy according to the data distribution is important**. We greatly appreciate the reviewer for pointing this out, as it highlights the broader generality of our method: the core idea of **"high-quality initialization (depending on data distribution) + our density-aware correction module"** is fully effective. We have added these results and the discussion on initialization strategies to the Appendix C.5 of the revised paper  for completeness, and we thank the reviewer again for the constructive feedback, which has strengthened the completeness of our work.
>
>
> **Question 2 & Weakness 1(e): Performance on tabular data.**
>
> **Response**:  Our additional experiments on **CNAE-9** and **Semeion**, confirm that our method also works well on tabular data. The results in Table Q2 show that our method improves SCAN [R1] on both datasets, with a more significant gain on CNAE-9. This indicates that the core idea of our method is not limited to image data and can also work on other types of data, such as tabular data. We thank the reviewer for raising this important question, which allows us to demonstrate the broader applicability of our approach. The results and additional analysis have been added to the Appendix C.8.
>
> [R1] SCAN: learning to classify images without labels. ECCV, 2020.
>
> **Table Q2: Performance of our method on tabular datasets (CNAE-9 and Semeion).**
> |           | **CNAE-9** |          |          | **Semeion** |          |          |
> | --------- | ---------- | -------- | -------- | ----------- | -------- | -------- |
> |           | ACC        | NMI      | ARI      | ACC         | NMI      | ARI      |
> | SCAN      | 55.9       | 43.6     | 34.8     | 54.0        | 46.8     | 33.4     |
> | SCAN+Ours | **63.9**   | **51.7** | **43.3** | **55.8**    | **52.3** | **40.8** |

---

> ### Author Response · Authors · 2025-11-22
>
> **Question 3 & Weakness 2: How is your sampling strategy related to Active Learning strategies? What could we learn from there, what are similarities?**
>
> **Response**: Active learning aims to improve model performance when the labeling budget is limited. Its key idea is the sampling strategy: choosing the most informative unlabeled samples so that each selected sample brings the most benefit to the model.
>
> Our sampling idea is similar to active learning. Both methods try to focus on samples that are most valuable or most informative for the model. Like uncertainty-based sampling in active learning, our method also pays more attention to samples that the model has not learned well, while ignoring samples that are redundant or already well learned. This helps improve training efficiency and clustering quality. The main difference is that active learning requires human labels and is used in supervised settings. In contrast, our sampling works fully in an unsupervised/self-supervised setting, without any manual labeling. It selects and filters samples based on model predictions and the feature distribution. To the best of our knowledge, **this is the first work that introduces a sample selection mechanism into deep clustering.**
>
> Experience from active learning shows that choosing the most informative samples can noticeably improve model performance. This supports our sampling module and suggests that ideas like diversity sampling or density-based sampling may help further improve our method. Thank you for this comment, **we have added a discussion in Section 2 of the revised paper.**
>
>
>
> **Weakness 1(a): Experimental evaluation on the effect of model complexity.**
>
> **Response**:  Our method is robust to different model complexities. To address this question, we conducted experiments on **ResNet-18** for CIFAR-10 and STL-10, and the results are shown in Table Q3(the results in Table 1 of the original paper use ResNet-34). We can see that our method consistently improves the clustering performance of CDC and SCAN on both the smaller ResNet-18 and the larger ResNet-34. This demonstrates that our method is robust to different model complexities.
>
> **Table Q3: Effect of model complexity on clustering performance (ResNet-18 vs. ResNet-34) for CIFAR-10 and STL-10.**
> |               | **CIFAR-10** |          |          | **STL-10** |          |          |
> | :-----------: | :----------: | :------: | :------: | :--------: | :------: | :------: |
> | **ResNet-18** |      ACC     |    NMI   |    ARI   |     ACC    |    NMI   |    ARI   |
> |      CDC      |     91.1     |   83.3   |   82.3   |    74.1    |   70.2   |   62.8   |
> |    CDC+Ours   |   **91.5**   | **83.8** | **83.0** |  **74.4**  | **71.3** | **64.0** |
> |      SCAN     |     82.9     |   77.8   |   72.5   |    86.4    |   76.2   |   73.8   |
> |   SCAN+Ours   |   **84.6**   | **78.2** | **73.5** |  **87.5**  | **77.6** | **75.4** |
> |               |              |          |          |            |          |          |
> | **ResNet-34** |      ACC     |    NMI   |    ARI   |     ACC    |    NMI   |    ARI   |
> |      CDC      |     94.2     |   88.1   |   88.1   |    93.0    |   85.8   |   85.6   |
> |    CDC+Ours   |   **94.7**   | **88.7** | **89.0** |  **93.6**  | **86.9** | **86.8** |
> |      SCAN     |     90.2     |   83.7   |   81.0   |    91.4    |   83.4   |   82.6   |
> |   SCAN+Ours   |   **92.2**   | **85.6** | **84.5** |  **92.7**  | **85.1** | **84.9** |

---

> ### Author Response · Authors · 2025-11-22
>
> **Weakness 1(b): Experimental evaluation on the effect of dataset size and number.**
>
> **Response**:  Our results show that even when the dataset becomes smaller, our method remains stable and consistently improves the baseline. We conduct experiments on the **CIFAR-10** dataset using different dataset sizes (60k, 50k, 40k) to study how stable and effective our method is when the amount of data changes. The results are shown in Table Q4.
>
> We find that across all dataset sizes, our method (CDC + Ours) brings consistent and clear improvements over the baseline CDC. This shows that our method is robust and works well even when the dataset becomes smaller. While the baseline CDC experiences a noticeable performance drop as the dataset size decreases, our method remains stable and maintains high performance across all settings.
>
> **Table Q4: Effect of dataset size on CDC performance on CIFAR-10.**
> | Dataset size | Method   | ACC      | NMI      | ARI      |
> | ------------ | -------- | -------- | -------- | -------- |
> | 60000        | CDC      | 94.2     | 88.1     | 88.1     |
> |              | CDC+Ours | **94.7** | **88.7** | **89.0** |
> | 50000        | CDC      | 93.4     | 86.4     | 85.8     |
> |              | CDC+Ours | **94.5** | **88.4** | **88.6** |
> | 40000        | CDC      | 92.7     | 86.5     | 85.3     |
> |              | CDC+Ours | **94.5** | **88.4** | **88.6** |
>
>
> **Weakness 1(c): Experimental evaluation on the effect of number of concepts per class/cluster.**
>
> **Response**:  The datasets we use naturally cover different levels of intra-class semantic complexity. For example, each class in CIFAR-10 represents a single semantic concept (although samples still show different visual patterns). In contrast, each superclass in CIFAR-20 is composed of five CIFAR-100 subclasses, so the semantic diversity within a class is much richer.
>
> Even though CIFAR-20 contains more concepts per class and has more complex structures, our method achieves stable and consistent improvements on both datasets. This shows that our approach is not sensitive to the number of concepts inside a class, and it can still perform well even when the class has richer and more diverse semantics. Therefore, our method is robust to changes in the number of concepts per class.

---

> ### Author Response · Authors · 2025-11-22
>
> **Weakness 3: How much runtime do the computations for the sampling need? Is the runtime acceleration dependent on properties of the data that can be predicted?**
>
> **Response**:  Our Dynamic Sample Selection (DSS) module introduces only a very light computational overhead, while in practice it reduces the total training time because it dynamically filters out well-learned samples and trains on fewer samples in later epochs. Table Q5 reports the additional time of our method (CDC + Ours) on CIFAR-10 and STL-10. On STL-10, DSS adds 1.5 s per epoch (\~6.3% of the CDC baseline), and on CIFAR-10 it adds 8.4 s per epoch (\~3.6% of the baseline).
>
> Based on our current experiments, the runtime acceleration mainly comes from removing stable samples during training. We do not find evidence that the acceleration depends on any specific or predictable property of the dataset. Therefore, we believe the acceleration is driven by the model's learning dynamics rather than by data characteristics: as training goes on, more samples become stable and are removed, naturally reducing the training cost.
>
> **Table Q5. Runtime and acceleration effect of DSS on CIFAR-10 and STL-10.**
> | **Time usage per epoch** | **STL-10** | **CIFAR-10** |
> | ------------------------ | ---------- | ------------ |
> | CDC                      | 23.9       | 232.4        |
> | DSS                      | 1.5        | 8.4          |
> | CDC+Ours                 | 19.2       | 185.8        |
>
> **Weakness 4: Using k-Means as initlal clustering should be discussed more. It is not clear to me how the value of k for the initial k-Means clustering was chosen. What happens if the data does not follow typical assumptions fitting k-Means.**
>
> **Response**:  In deep clustering tasks, the true number of classes is typically known, which is standard practice in this research area. Therefore, we simply set the value of k in the K-Means initialization to the true number of classes.
>
> As discussed earlier in **Question 1 & Weakness 1(d)**, K-Means initialization does not work equally well for all data distributions. For example, on COIL-20, K-Means performs poorly due to the non-convex and highly structured data manifold. In contrast, spectral clustering provides a much better initialization, and adding our module on top of spectral clustering further improves performance. These results show that **choosing an initialization strategy that matches the data distribution is important**. We appreciate the reviewer for raising this point, as it highlights the generality of our method: the core idea of **"high-quality initialization (chosen according to data characteristics) + our density-aware correction module"** works well across different distributions. We have added the COIL-20 results and a discussion on initialization strategies in Appendix C.5 of the revised paper.
>
>
> **Response to Minor Stuff.**
>
> Thank you for pointing out these issues. We have corrected the typo in line 90 and moved Table 3 to proper location.

---

> ### Comment · Reviewer_37QQ · 2025-11-25
> **Answer to Rebuttal**
>
> Thanks to the authors for their extensive elaborations and additional experiments! I'm especially happy to see the improvement of results on the COIL 20 dataset that another initialization can achieve. You addressed my concerns and I'll keep my recommendation for acceptance.

---

> > ### Author Response · Authors · 2025-11-25
> > **We are glad that your concerns have been addressed.**
> >
> > Dear Reviewer **37QQ**,
> >
> > **We are glad that your concerns have been addressed**. Your valuable comments are greatly important for improving the quality of this paper. Thanks again for your supportive comments.
> >
> > Regards from the authors.

---

### Official Review · Reviewer_WmJa · 2025-10-31

**Soundness:** 2
**Presentation:** 3
**Contribution:** 3
**Rating:** 4
**Confidence:** 4

**Summary:**

The paper proposes improvements to the self-labeling stage of deep clustering. First, it introduces a density-aware clustering head initialization that downweights redundant high-density samples and upweights rare/low-density samples when forming prototypes. Second, it proposes a dynamic sample selection strategy that prunes stable samples based on pseudo-label stability and consistency between weakly and strongly augmented views, allowing training to focus on unstable or underlearned samples. Overall, this leads to improved clustering results across several standard image clustering benchmarks and results in better training efficiency, since fewer samples are actively optimized in later epochs.

**Strengths:**

- The paper targets an important stage used in state of the art deep clustering algorithms: self-labeling. The motivation is clearly laid out and depicts a practical issue: overfitting to dense/easy samples while underfitting rare samples.
- The method improves both clustering accuracy and efficiency across several standard datasets and multiple baselines, suggesting it is broadly useful. The authors also provide a hyperparameter study on (
α,
k,
ϵ) and show that the approach is generally robust to
α and k.

**Weaknesses:**

- Training is always stopped after a fixed 100 epochs. Unsupervised stopping criteria remain an open problem in the self-labeling stage. The proposed framework (especially with dynamic pruning) could, in principle, offer a stopping signal, but this is not analyzed.
- The evaluation does not clearly report model selection details or variance across seeds. Because self-labeling inherently involves noise and instability (due to training on pseudo-labels), averaged results and standard deviations are important to properly interpret the reported gains.

**Questions:**

- How are models selected and tuned? What are the standard deviations across runs, e.g. for 5 to 10 seeds. How stable is the method overall?
- You run for 100 epochs, but given that you explicitly track stability and prune stable samples, could those same signals (e.g., the fraction of samples no longer changing) be used as a stopping heuristic?
- What happens if you continue training far beyond 100 epochs (e.g., 500+)? Do you eventually prune almost all samples, or does the model start overfitting the small subset of rare or ambiguous samples that never get pruned?
- At the moment a sample is pruned as stable, how often is it actually assigned the correct ground-truth class? Showing that would address the concern that you might be confidently pruning wrong assignments and locking in errors. That could, however, have the benefit of reducing error propagation by no longer learning on those errors.

---

> ### Author Response · Authors · 2025-11-22
>
> Thank you for your valuable comments.
>
> ------
>
> **Question 1 & Weakness 2: How are models selected and tuned? What are the standard deviations across runs. How stable is the method overall?**
>
> **Response**: We exactly follow the same experimental settings as CDC [R1], and **all training details are provided in Appendix B of the original paper**. Although our Density-Aware Clustering Head Initialization(DACHI) module introduces two parameters (α and k), Figs. 4(b)(c) of the original paper show that both of them are insensitive across datasets. For the Dynamic Sample Selection(DSS) module, the main hyperparameter is ϵ. The default value ($1e^{-1}$) works well for almost all methods and datasets, and **we do not perform any dataset-specific tuning**. Only in one special case (CC [R2] on STL-10 and ImageNet-10), we use a smaller value ($1e^{-2}$). This is because CC uses instance-level strong/weak contrastive learning, which naturally gives higher consistency, causing a higher pruning rate under the same threshold. In addition, STL-10 and ImageNet-10 have relatively few samples (13K), and a high pruning rate may lead to instability. Using a smaller ϵ simply keeps the pruning ratio similar to the other methods, and is not the result of fine-grained tuning.
>
> [R1] Towards calibrated deep clustering network. ICLR, 2025.
>
> [R2] Contrastive clustering. AAAI, 2021.
>
> The experimental results in Table 1 of the original paper were obtained from a single run under a random seed of 5.  To further analyze the robust performance of our method, we conducted five runs on CIFAR-10, CIFAR-20, and STL-10, with different random seeds, and the results are presented in the Table Q1. It can be observed that **our method has higher clustering peroformance under all random seeds, suggesting its excellent stability**. Moreover, the corresponding results and analysis have been added in Appendix C.6 of the paper.
>
> **Table Q1: The clustering performance ACC, NMI, ARI (mean±std %) on CIFAR-10, CIFAR-20 and STL-10.**
>
> |           | **CIFAR-10** |              |              | **CIFAR-20** |              |              |  **STL-10**  |              |              |
> | :-------: | :----------: | :----------: | :----------: | :----------: | :----------: | :----------: | :----------: | :----------: | :----------: |
> |           |     ACC      |     NMI      |     ARI      |     ACC      |     NMI      |     ARI      |     ACC      |     NMI      |     ARI      |
> |   SCAN    |   88.6±1.9   |   83.2±0.3   |   79.4±1.3   |   51.0±1.0   |   53.9±0.9   |   37.1±0.7   |   91.4±0.5   |   83.5±0.6   |   82.7±0.8   |
> | SCAN+Ours | **91.8±0.5** | **85.0±0.6** | **83.7±1.0** | **55.3±0.4** | **56.7±0.5** | **40.5±0.7** | **92.5±0.1** | **85.0±0.3** | **84.6±0.3** |
> |    CDC    |   94.1±0.3   |   87.9±0.4   |   87.8±0.6   |   61.7±0.3   |   61.1±0.2   |   46.0±0.2   |   93.0±0.1   |   85.9±0.1   |   85.6±0.1   |
> | CDC+Ours  | **94.6±0.2** | **88.5±0.2** | **88.7±0.4** | **62.4±0.3** | **61.4±0.1** | **46.3±0.3** | **93.5±0.1** | **86.7±0.2** | **86.6±0.3** |

---

> ### Author Response · Authors · 2025-11-22
>
> **Question 2 & Weakness 1: Could those same signals (e.g., the fraction of samples no longer changing) be used as a stopping heuristic?**
>
> **Response**:  We train all methods for 100 epochs primarily for two reasons:
>
> **1. Fair comparison.** To ensure a fair evaluation against baseline methods, we keep the number of training epochs fixed across all approaches.
>
> **2. Lack of universal early stopping criteria.** Deep clustering methods generally do not have widely accepted or universal stopping rules. As a result, using a fixed number of epochs remains standard practice in this field.
>
> Fortunately, we additionally conduct experiments and find that **our sample stability assessment can indeed serve as a practical heuristic for early stopping**. Empirically, as training progresses, the number of samples the model has "mastered" and thus temporarily removes increases, and this growth gradually slows and eventually saturates. Based on this observation, we design a simple stopping criterion: we track the number of samples removed at each epoch and record the maximum removed count. When this maximum does not change for the most recent K epochs, we stop training.
>
> We evaluate our proposed stopping strategy in combination with CDC on CIFAR-10 and STL-10, testing several values of K(10, 20, 30, 40). We report the stopping time, the performance at the stopping point, and the best achievable performance. As shown in Table Q2, for K = 10, 20, the performance at stopping is within 1% of the best performance. For K = 30, 40, the stopped model achieves performance almost identical to the optimal result. This shows that our stopping strategy can find a point that is very close to the best result without training for all epochs. This highlights the practical benefit of our method. The related results and analysis have been added to Appendix C.7 of the revised paper.
>
> **Table Q2: Performance of the sample stability based early stopping strategy on CIFAR-10 and STL-10.**
>
> |           | **CIFAR-10** |      |      |      | **STL-10** |      |      |      |
> | :-------: | :----------: | :--: | :--: | :--: | :--------: | :--: | :--: | :--: |
> |           |  Stop epoch  | ACC  | NMI  | ARI  | Stop epoch | ACC  | NMI  | ARI  |
> | EPOCH=100 |      —       | 94.4 | 88.3 | 88.5 |     —      | 93.4 | 86.5 | 86.2 |
> |   Best    |      —       | 94.6 | 88.5 | 88.7 |     —      | 93.5 | 86.6 | 86.7 |
> |   K=10    |      49      | 94.0 | 87.2 | 87.4 |     59     | 93.0 | 85.8 | 85.5 |
> |   K=20    |      69      | 94.3 | 87.9 | 88.2 |     77     | 93.1 | 85.9 | 85.7 |
> |   K=30    |      69      | 94.3 | 87.9 | 88.2 |     98     | 93.4 | 86.6 | 86.4 |
> |   K=40    |      —       |  —   |  —   |  —   |     98     | 93.4 | 86.6 | 86.4 |
>
>
>
> **Question 3: What happens if you continue training far beyond 100 epochs (e.g., 500+)?**
>
> **Response**: **Our method will not prune almost all samples**. Specifically, our sampling mechanism is dynamic: we continually assess each sample's learning state, and the removed samples are not discarded permanently. When they become unstable, they are added back into training. As a result, our method never discards all samples permanently, and after enough training, the number of removed samples naturally stabilizes.
>
> We trained the model for 1,000 epochs on CIFAR-10 and STL-10 to examine this behavior. As shown in Table Q3, the number of removed samples after 1,000 epochs is only about **1%** higher than after 100 epochs, while the extended training time leads to improved performance.
>
> **Table Q3: Effect of extended training (100 vs. 1000 epochs) on pruning behavior and clustering performance.**
>
> | Epoch | **CIFAR-10** |      |      |      | **STL-10** |      |      |      |
> | ----- | ------------ | ---- | ---- | ---- | ---------- | ---- | ---- | ---- |
> |       | Pruned/All   | ACC  | NMI  | ARI  | Pruned/All | ACC  | NMI  | ARI  |
> | 100   | 10602/60000  | 94.4 | 88.3 | 88.5 | 2848/13000 | 93.4 | 86.5 | 86.2 |
> | 1000  | 11184/60000  | 95.4 | 90.0 | 90.0 | 3038/13000 | 93.5 | 86.5 | 86.4 |

---

> ### Author Response · Authors · 2025-11-22
>
> **Question 4: At the moment a sample is pruned as stable, how often is it actually assigned the correct ground-truth class?**
>
> **Response**: **The samples pruned as "stable" show very high accuracy, significantly higher than the accuracy over all samples.** We evaluated CDC+Ours on CIFAR-10 and STL-10, comparing the per-epoch accuracy of removed samples with the accuracy over all samples, as shown in Table Q4. The results show that removed samples consistently have higher accuracy at each stage, indicating that the model usually makes correct predictions when marking samples as "stable". The full training curves in Appendix C.3 further show that this trend holds across all epochs.
>
> **Table Q4: Accuracy of pruned samples versus all samples during training on CIFAR-10 and STL-10.**
> | Epoch        | 10   |        | 30   |        | 50   |        | 70   |        | 100  |        |
> | ------------ | ---- | ------ | ---- | ------ | ---- | ------ | ---- | ------ | ---- | ------ |
> |              | All  | Pruned | All  | Pruned | All  | Pruned | All  | Pruned | All  | Pruned |
> | **CIFAR-10** | 93.1 | 96.2   | 93.8 | 96.6   | 94.2 | 96.9   | 94.3 | 96.7   | 94.5 | 97.1   |
> | **STL-10**   | 92.2 | 95.2   | 92.7 | 95.1   | 93.0 | 95.2   | 93.1 | 95.2   | 93.4 | 95.4   |

---

> > ### Comment · Reviewer_WmJa · 2025-11-25
> >
> > Thank you for the detailed responses. The additional experiments, particularly the early stopping analysis, strengthen the paper in my view, therefore I am raising my score from 4 to 6.

---

> > > ### Author Response · Authors · 2025-11-25
> > >
> > > Dear Reviewer **WmJa**,
> > >
> > >
> > > Thanks for your kind reply. **We are happy that your concerns have been addressed**. Thanks for raising your score.
> > >
> > >
> > > Regards from the authors.

---

### Author Response · Authors · 2025-11-25
**Looking forward to your further comments**

Dear **Reviewers**,

Thank you for taking the time to review our manuscript and for your valuable feedback and recognition. We have carefully addressed all the comments and concerns raised, as reflected in our detailed responses and the revised manuscript and supplementary material.

We sincerely appreciate your efforts and look forward to your further assessment.

Best regards,

The Authors

---

### Author Response · Authors · 2025-12-03
**Global Response and Revisions Summary**

Dear Reviewers, Area Chairs, and Program Chairs,

We sincerely thank the reviewers for their thoughtful comments and valuable questions, which have significantly strengthened our paper. We also deeply appreciate the Area Chair’s time and careful handling of the review process.

To support your assessment of our submission, we offer a brief summary of the review process and the improvements made during the rebuttal phase. We would also like to clarify that all discussions and exchanges occurred before the November 27 review-leak incident.

### **Summary of Rebuttal**

**Reviewer WmJa (Original Score: 4 → Raised Score: 6):**
Reviewer WmJa agreed that our method has a **clear motivation** and **broad applicability**, but initially expressed concerns about training stability, stopping strategies, and the reliability of sample selection. After we provided results with multiple random seeds, an early stopping scheme, long-training experiments, and an accuracy analysis of the removed samples, these concerns were resolved. Reviewer WmJa confirmed that the additional experiments, particularly the early stopping analysis, **strengthen the paper**,  and therefore **raised the score from 4 to 6**.

**Reviewer 37QQ (Original Score: 8):**
Reviewer 37QQ highly valued the **plug-and-play** nature of our sample selection method, its simplicity, and its improvement to training efficiency. In response to the concern about "initialization sensitivity", we added experiments using spectral clustering–based initialization on COIL-20, showing that our method provides significant gains under high-quality initialization. We also included additional experiments on tabular datasets to demonstrate the broad applicability of our method. Reviewer 37QQ confirmed that the new results **addressed the concerns** and **maintained a positive recommendation**.

**Reviewer 8obk (Original Score: 4):**
Reviewer 8obk acknowledged the **strong adaptability** of our method. The main initial concerns were the reliability of sample selection and the applicability of our approach to non-image modalities. We provided detailed experimental analysis showing that our constraints are highly reliable, and our experiments on non-image datasets further showed strong cross-modality generalization. The reviewer **accepted these new results** and suggested testing on more complex datasets. In response, we added experiments using different model architectures on the more challenging Tiny-ImageNet dataset, demonstrating the consistent effectiveness of our method and addressing the reviewer’s concerns. We believe that, had the rebuttal phase proceeded normally, Reviewer 8obk would have raised the score.

**Reviewer K4YF (Original Score: 4):**
Reviewer K4YF acknowledged the **novelty** and **broad applicability** of our method. The concerns focused on the computational and memory overhead of the DACHI and DSS modules, as well as their interaction. We provided a detailed breakdown of time and memory usage, showing that both DACHI and DSS are lightweight. We also compared different initialization strategies and analyzed their effects on the number of samples pruned by DSS. The results show that DACHI initialization prunes a similar number of samples as CDC initialization but achieves clearly better performance. Due to the incident that occurred on November 27, the reviewer did not have the opportunity to respond. Nevertheless, we believe that we have addressed the majority of Reviewer K4YF’s concerns.

---

> ### Author Response · Authors · 2025-12-03
>
> ### **Overall Assessment and Summary of Revisions**
>
> We thank all the reviewers for their valuable efforts in reviewing our paper and for providing insightful feedback.
>
> Overall, the reviewers acknowledged the following strengths of our work:
>
> (1) Addressing the challenge of **overfitting** to high-density samples and **underfitting** to low-density samples in deep clustering (**WmJa, 8obk**).
>
> (2) Introducing a **novel** sampling strategy that can be integrated into various deep clustering methods (**37QQ, 8obk, K4YF**).
>
> (3) Delivering **consistent gains in both clustering accuracy and efficiency** across several datasets and baselines (**WmJa, 37QQ, K4YF**).
>
> (4) Demonstrating **robustness to hyperparameters** (**WmJa**).
>
> (5) Providing a **simple and intuitive** method that is easy to implement (**37QQ**).
>
> (6) Confirming contributions of both DACHI and DSS through ablation studies (**K4YF**).
>
>
>
> Meanwhile, we have substantially **expanded and strengthened** the paper by **adding several new analyses and experiments**, addressing all reviewer concerns. The major additions include:
>
> (1) **Accuracy of stable samples (Appendix C.3):** We newly provide a comparison between the accuracy of stable samples and the accuracy of all samples.
>
> (2) **Sample visualization (Appendix C.4):** We newly include visualizations of both stable and unstable samples.
>
> (3) **Initialization strategy discussion (Appendix C.5):** We add a new analysis comparing different initialization strategies.
>
> (4) **Method stability (Appendix C.6):** We newly provide multi-run results under different random seeds.
>
> (5) **Early stopping strategy (Appendix C.7):** Based on the stability signal of samples, we propose a feasible stopping criterion for training.
>
> (6) **Performance on non-image datasets (Appendix C.8):** We additionally evaluate our method on non-image datasets to demonstrate broader applicability.
>
> (7) **Active learning background (Section 2):** We add more background on active learning.
>
> In addition, we **provide further clarification** in the rebuttal regarding:
>
>  – **System resource usage**, including a detailed discussion of time and memory costs;
>
>  – **Additional experimental insights**, including the effects of model architecture, training data size, training duration, encoder quality, and the interaction between our two modules.

---

### Meta-Review · Area_Chair_jrk5 · 2026-01-07

**Summary:**

Almost all of the reviewers' concerns were well addressed and the reviewers would likely have converged to a borderline accept decision for the paper based on its clear motivation, simple plug-and-play method that improves multiple strong baselines, and the obtained consistent gains, accompanied by detailed insights into compute and memory demands as well as generalization capabilities to different data sets.

**Reviewer Concerns:**

Reviewer WmJa
- Lack of unsupervised stopping criterion; fixed 100-epoch training.
  - Addressed: authors proposed a stability-based early stopping heuristic (varying K) with results near the best performance
- Missing details on model selection and stability/variance across seeds
  - Addressed: multi-seed results added (5 runs), showing robustness
- Behavior with long training (500–1000+ epochs)
  - Addressed: 1000-epoch experiments show modest further pruning and improved performance
- Reliability of pruning decisions (are "stable" samples correctly labeled?)
  - Addressed: per-epoch accuracy shows pruned samples have higher accuracy than overall

Reviewer 37QQ
- Depth of evaluation: dependence on model complexity, dataset size, and number of concepts per class
  - Addressed: added experiments across ResNet-18/34, varying CIFAR-10 sizes, and CIFAR-20 vs CIFAR-10
- Performance on consistent-density data (e.g., COIL-20)
  - Addressed: method improved results
- Performance on tabular data
  - Addressed: added CNAE-9 and Semeion experiments; showed performance improvements
- Relation to Active Learning
  - Addressed: added discussion connecting to AL strategies; reviewer satisfied.
- Runtime/acceleration details and predictability
  -  Addressed: overhead quantified; acceleration attributed to pruning dynamics, not specific data properties
- k-means initialization choice of $k$ and handling non-k-means-friendly data
  - Addressed: $k$ set to true class count;

Reviewer 8obk
- Dependence on pre-trained encoder quality; handling weaker or non-pretrained models
  - Partially addressed: experiments with weaker ResNet-18 pretraining and added Tiny-ImageNet; authors note pretraining is standard
- Larger-scale validation:
  - Partially addressed: authors supplemented the results but the reviewer would prefer ImageNet-scale complexity
- Early-stage consistency/pseudo-label reliability
  - Addressed: constraints require persistent consistency over multiple epochs; pruned samples consistently more accurate
- Cross-modality generalization:
  - Addressed: added non-image datasets (text/tabular) with gains.
- Computational overhead
  - Addressed: added time/memory breakdown

Reviewer K4YF
- DSS and DACHI computational/memory overhead; wall-clock analysis
  - Addressed: per-epoch time breakdown and memory usage; DACHI initialization overhead measured and small
- Sensitivity of DSS pruning threshold $\epsilon$
  - Addressed: default $\epsilon$ works broadly; rationale for smaller $\epsilon$ in CC on small datasets; trade-off guidance between speed and performance
- Interplay between DACHI and DSS
  -  Addressed: comparison of initialization strategies and $\alpha$ settings; DACHI and CDC prune similar counts; random initialization prunes more but performs worse
- Modest gains on strong baseline (CDC)
  - Addressed: clarified +1.1% average gain on SOTA is meaningful; consistent improvements across multiple baselines.

**Reviewer Scores:**

* Reviewer WmJa's concerns were well addressed and they would have increased the score to borderline accept
* Reviewer 378Q's rating of the paper was already very positive and they would have maintained the score
* Reviewer 8obk might have increased their score (-> borderline accept)
* Reviewer K4YF's concerns were well addressed and they might have increased their score (-> borderline accept)

---

### Decision · Program_Chairs · 2026-01-26

Accept (Poster)